# Optimal Learning Rates for Regularized Conditional Mean Embedding

**Zhu Li**[*]
Gatsby Computational Neuroscience Unit
University College London
zhu.li@ucl.ac.uk

**Dimitri Meunier**[*]
Gatsby Computational Neuroscience Unit
University College London
dimitri.meunier.21@ucl.ac.uk

**Mattes Mollenhauer**
Department of Mathematics and Computer Science
Freie Universität Berlin
mattes.mollenhauer@fu-berlin.de

**Arthur Gretton**
Gatsby Computational Neuroscience Unit
University College London
arthur.gretton@gmail.com

## Abstract

We address the consistency of a kernel ridge regression estimate of the conditional mean embedding (CME), which is an embedding of the conditional distribution of $Y$ given $X$ into a target reproducing kernel Hilbert space $\mathcal{H}_Y$. The CME allows us to take conditional expectations of target RKHS functions, and has been employed in nonparametric causal and Bayesian inference. We address the misspecified setting, where the target CME is in the space of Hilbert-Schmidt operators acting from an input interpolation space between $\mathcal{H}_X$ and $L_2$, to $\mathcal{H}_Y$. This space of operators is shown to be isomorphic to a newly defined vector-valued interpolation space. Using this isomorphism, we derive a novel and adaptive statistical learning rate for the empirical CME estimator under the misspecified setting. Our analysis reveals that our rates match the optimal $O(\log n/n)$ rates without assuming $\mathcal{H}_Y$ to be finite dimensional. We further establish a lower bound on the learning rate, which shows that the obtained upper bound is optimal.

## 1 Introduction

Approximation of the conditional expectation operator is a central issue in the statistical learning community, and many approaches have been proposed [42, 20, 21, 22]. Given random variables $X$ and $Y$, the conditional expectation operator for a function $f$ is defined

$$[Pf](x) := \mathbb{E}[f(Y)|X = x].$$

Conventional parametric models to approximate $P$ often involve density estimation and expensive numerical analysis. Hence, recent studies attempt to explore a new framework to approximate $P$ via kernel methods. Specifically, given kernels $k_X$ and $k_Y$ with corresponding reproducing kernel Hilbert space $\mathcal{H}_X$ and $\mathcal{H}_Y$ for $X$ and $Y$ respectively, we may define the conditional mean embedding (CME) as $F_*(x) := \mathbb{E}[k_Y(\cdot, Y)|X = x]$, and we may employ the reproducing property to obtain $[Pf](x) = \langle f, F_*(x) \rangle_{\mathcal{H}_Y}$ for any $f \in \mathcal{H}_Y$. The advantage of the CME framework is that it allows the straightforward evaluation of conditional expectations of any function in $\mathcal{H}_Y$. The CME framework has been applied successfully to many learning problems such as probabilistic inference [35], reinforcement learning [28, 16] and causal inference [25, 31].

---

[*] denotes equal contribution.

Despite these successful applications, there have been two main challenges in establishing a rigorous theory of CMEs. The first challenge, remarkably, has been in establishing a principled and sufficiently general definition of the conditional mean embedding itself. The CME was originally introduced as an operator mapping from $\mathcal{H}_X$ to $\mathcal{H}_Y$ [12, 37]. This definition has the benefit of elegance, and of a straightforward expression in terms of feature covariances and cross-covariances. A disadvantage is that the definition requires the conditional mean $\mathbb{E}[g(Y)|X = \cdot] \in \mathcal{H}_X, \forall g \in \mathcal{H}_Y$. This strong assumption may be violated in practice (see [18, 19] and [13, Section 3.1] for illustrations and alternative requirements), and significantly restricts the class of distributions on which we can define a CME.

An alternative approach, due to [15], is to express the conditional mean embedding as the solution of a least-squares regression problem in a vector-valued RKHS [5, 6]. In subsequent work, a rigorous measure-theoretic definition of the conditional mean embedding as the $\mathcal{H}_Y$-valued square integrable function $F_*$ is established in [29, 18], which is the definition we will use in the present work. Both [15, 29] connect this CME definition to the original operator-mapping definition by means of a surrogate loss, which upper bounds the regression loss. A direct connection remained elusive until the work of [26, 19], which show that under denseness assumptions, the CME can be arbitrarily well approximated by a Hilbert-Schmidt operator from $\mathcal{H}_X$ to $\mathcal{H}_Y$, thus connecting the operator-theoretic and measure-theoretic definitions.

The second challenge has been in obtaining consistency results and the optimal learning rates for empirical estimates of the CME. An early consistency analysis of the sample estimator, due to [36], requires very strong smoothness assumptions. A more refined analysis, due to [15], attains the minimax optimal learning rate $O(\log n/n)$ for the sample estimator, but only in the case where $\mathcal{H}_Y$ is finite dimensional. For the infinite dimensional RKHS, [31] and [29] establish consistency in the well-specified case, with learning rates of $O(n^{-1/6})$ and $O(n^{-1/4})$. Nevertheless, the obtained rates are far from optimal and consistency under misspecified setting was not established. Recently, [41] obtains a sharper rate under the misspecified case using the interpolation RKHS. The results of [41] impose assumptions, however, which strongly limit their applicability (refer to Remark 5 for a rigorous discussion):

1. They require an explicit relation between the smoothness of the target CME and the size of the RKHS. In particular, when the kernel has slow eigenvalue decay (as in the case of Matérn kernels, for example), the setting is very close to the well-specified scenario.

2. They rely on the explicit construction of an interpolation RKHS. Unlike in [11], where a similar approach is based only on equivalence classes of functions (i.e., Sobolev-like spaces), this concept requires the embedding of the RKHS into the corresponding $L^2$-space (or equivalently the integral operator) to be injective—which is generally not the case (see [40] for details). Counterexamples can easily be constructed when one considers degenerate pushforward measures on the RKHS in one or more coordinate directions (for example point masses). By contrast, the authors of [11] do not explicitly require the injectivity in the real-valued learning scenario. Moreover, in case where the chosen kernel has slow eigenvalue decay, the constructed interpolation RKHS is not well-defined.

Finally, to our knowledge, there is presently no result establishing a matching lower bound for the CME learning rate in the case where $\mathcal{H}_Y$ may be infinite dimensional. Hence, whether the obtained upper rate is optimal remains unknown.

In the present work, we address the challenges mentioned above. Building on [29, 26] and the interpolation space theory results of [40, 11], we introduce an **interpolation space consisting of vector-valued functions** via a natural tensor product construction. This concept is compatible with the recent measure-theoretic definition of the CME due to [29] and allows to prove convergence in the misspecified setting without the limitations of prior work. Based on this novel vector-valued interpolation space, we establish **consistency and convergence rates of the CME sample estimator in the misspecified setting**. In particular, under certain benign conditions, we obtain the optimal $O(1/n)$ learning rate up to a logarithmic factor. This matches with the current optimal analysis from [15] without the restrictive assumption of finite dimensional $\mathcal{H}_Y$. Thanks to our operator-theoretic definition of the CME, and unlike [41], we do not require an a-priori relation between the rate of kernel eigenvalue decay and the smoothness of the conditional mean operator (i.e., our results apply generally in the misspecified setting). Finally, in Theorem 3, we provide a novel **lower bound on the**

**CME learning rate**, which demonstrates that the obtained upper rate is optimal in the setting of a smooth CME operator.

## 2  Background

Throughout the paper, we consider two random variables $X, Y$ defined respectively on the second countable locally compact Hausdorff spaces $E_X$ and $E_Y$ endowed with their respective Borel $\sigma$-field $\mathcal{F}_{E_X}$ and $\mathcal{F}_{E_Y}$. We let $(\Omega, \mathcal{F}, \mathbb{P})$ be the underlying probability space with expectation operator $\mathbb{E}$. Let $\pi$ and $\nu$ be the pushforward of $\mathbb{P}$ under $X$ and $Y$ respectively, i.e., $X \sim \pi$ and $Y \sim \nu$. We use the Markov kernel $p : E_X \times \mathcal{F}_{E_Y} \to \mathbb{R}_+$ to define the conditional distribution:

$$\mathbb{P}[Y \in A | X = x] = \int_A p(x, dy),$$

for all $x \in E_X$ and events $A \in \mathcal{F}_{E_Y}$. We denote the space of real-valued Lebesgue square integrable functions on $(E_X, \mathcal{F}_{E_X})$ with respect to $\pi$ as $L_2(E_X, \mathcal{F}_{E_X}, \pi)$ abbreviated $L_2(\pi)$ and similarly for $\nu$ we use $L_2(E_Y, \mathcal{F}_{E_Y}, \nu)$ abbreviated $L_2(\nu)$. Let $B$ be a separable Banach space with norm $\| \cdot \|_B$ and $H$ a separable real Hilbert space with inner product $\langle \cdot, \cdot \rangle_H$. We write $\mathcal{L}(B, B')$ as the Banach space of bounded linear operators from $B$ to another Banach space $B'$, equipped with the operator norm $\| \cdot \|_{B \to B'}$. When $B = B'$, we simply write $\mathcal{L}(B)$ instead. We also let $L_p(E_X, \mathcal{F}_{E_X}, \pi; B)$, abbreviated $L_p(\pi; B)$, the space of strongly $\mathcal{F}_{E_X} - \mathcal{F}_B$ measurable and Bochner $p$-integrable functions $f : E_X \to B$ for $1 \le p \le \infty$. Finally, we denote the $p$-Schatten class $S_p(H, H')$ to be the space of all compact operators $C$ from $H$ to another Hilbert space $H'$ such that $\|C\|_{S_p(H,H')} := \left\| (\sigma_i(C))_{i \in J} \right\|_{\ell_p}$ is finite. Here $\| (\sigma_i(C))_{i \in J} \|_{\ell_p}$ is the $\ell_p$ sequence space norm of the sequence of the strictly positive singular values of $C$ indexed by the countable set $J$. For $p = 2$, $S_2(H, H')$ is the Hilbert space of Hilbert-Schmidt operators from $H$ to $H'$.

**Tensor Product of Hilbert Spaces ([1], Section 12):** Denote $H \otimes H'$ the tensor product of Hilbert spaces $H, H'$. The Hilbert space $H \otimes H'$ is the completion of the algebraic tensor product with respect to the norm induced by the inner product $\langle x_1 \otimes x_1', x_2 \otimes x_2' \rangle_{H \otimes H'} = \langle x_1, x_2 \rangle_H \langle x_1', x_2' \rangle_{H'}$ for $x_1, x_2 \in H$ and $x_1', x_2' \in H'$ defined on the elementary tensors of $H \otimes H'$. This definition extends to $\mathrm{span}\{x \otimes x' | x \in H, x' \in H'\}$ and finally to its completion. The space $H \otimes H'$ is separable whenever both $H$ and $H'$ are separable. The element $x \otimes x' \in H \otimes H'$ is treated as the linear rank-one operator $x \otimes x' : H' \to H$ defined by $y' \to \langle y', x' \rangle_{H'} x$ for $y' \in H'$. Based on this identification, the tensor product space $H \otimes H'$ is isometrically isomorphic to the space of Hilbert-Schmidt operators from $H'$ to $H$, i.e., $H \otimes H' \simeq S_2(H', H)$. We will hereafter not make the distinction between those two spaces and see them as identical. If $\{e_i\}_{i \in I}$ and $\{e_j'\}_{j \in J}$ are orthonormal basis in $H$ and $H'$, $\{e_i \otimes e_j'\}_{i \in I, j \in J}$ is an orthonormal basis in $H \otimes H'$.

**Remark 1** ([1], Theorem 12.6.1). *Consider the Bochner space $L_2(\pi; H)$ where $H$ is a separable Hilbert space. One can show that $L_2(\pi; H)$ is isometrically identified with the tensor product space $H \otimes L_2(\pi)$.*

**Reproducing Kernel Hilbert Spaces, Covariance Operators:** We let $k_X : E_X \times E_X \to \mathbb{R}$ be a symmetric and positive definite kernel function and $\mathcal{H}_X$ be a vector space of $E_X \to \mathbb{R}$ functions, endowed with a Hilbert space structure via an inner product $\langle \cdot, \cdot \rangle_{\mathcal{H}_X}$. $k_X$ is a reproducing kernel of $\mathcal{H}_X$ if and only if: 1. $\forall x \in E_X, k_X(\cdot, x) \in \mathcal{H}_X; 2. \forall x \in E_X$ and $\forall f \in \mathcal{H}_X, f(x) = \langle f, k_X(x, \cdot) \rangle_{\mathcal{H}_X}$. A space $\mathcal{H}_X$ which possesses a reproducing kernel is called a reproducing kernel Hilbert space (RKHS)[2]. We denote the canonical feature map of $\mathcal{H}_X$ as $\phi_X(x) = k_X(\cdot, x)$. Similarly for $E_Y$, we consider a RKHS $\mathcal{H}_Y$ with symmetric and positive definite kernel $k_Y : E_Y \times E_Y \to \mathbb{R}$ and canonical feature map denoted as $\phi_Y$.

We require some technical assumptions on the previously defined RKHSs and kernels:

1. $\mathcal{H}_X$ and $\mathcal{H}_Y$ are separable, this is satisfied if $E_X$ and $E_Y$ are Polish spaces and $k_X, k_Y$ are continuous [38];
2. $k_X(\cdot, x)$ and $k_Y(\cdot, y)$ are measurable for $\pi$-almost all $x \in E_X$ and $\nu$-almost all $y \in E_Y$;
3. $k_X(x, x) \le \kappa_X^2$ for $\pi$-almost all $x \in E_X$ and $k_Y(y, y) \le \kappa_Y^2$ for $\nu$-almost all $y \in E_Y$.

Note that the above assumptions are not restrictive in practice, as well-known kernels such as the Gaussian, Laplacian and Matérn kernels satisfy all of the above assumptions on $\mathbb{R}^d$. We now

introduce some facts about the interplay between $\mathcal{H}_X$ and $L_2(\pi)$, which has been extensively studied by [32, 33],[9] and [40]. We first define the (not necessarily injective) embedding $I_\pi : \mathcal{H}_X \to L_2(\pi)$, mapping a function $f \in \mathcal{H}_X$ to its $\pi$-equivalence class $[f]$. The embedding is a well-defined compact operator as long as its Hilbert-Schmidt norm is finite. In fact, this requirement is satisfied since its Hilbert-Schmidt norm can be computed as

$$\|I_\pi\|_{S_2(\mathcal{H}_X, L_2(\pi))} = \|k_X\|_{L_2(\pi)} := \left( \int_{E_X} k_X(x,x)\mathrm{d}\pi(x) \right)^{1/2} < \infty.$$

The adjoint operator $S_\pi := I_\pi^* : L_2(\pi) \to \mathcal{H}_X$ is an integral operator with respect to the kernel $k_X$, i.e. for $f \in L_2(\pi)$ and $x \in E_X$ we have

$$(S_\pi f)(x) = \int_{E_X} k_X(x,x') f(x') \, \mathrm{d}\pi(x')$$

Next, we define the self-adjoint and positive semi-definite integral operators

$$L_X := I_\pi S_\pi : L_2(\pi) \to L_2(\pi) \quad \text{and} \quad C_{XX} := S_\pi I_\pi : \mathcal{H}_X \to \mathcal{H}_X$$

These operators are trace class and their trace norms satisfy

$$\|L_X\|_{S_1(L_2(\pi))} = \|C_{XX}\|_{S_1(\mathcal{H}_X)} = \|I_\pi\|_{\mathcal{H}_X \to L_2(\pi)}^2 = \|S_\pi\|_{L_2(\pi) \to \mathcal{H}_X}^2.$$

**Vector-valued RKHS**  We also give a brief overview of the vector-valued reproducing kernel Hilbert space (vRKHS). We refer the reader to [5] and [6] for more detail.

**Definition 1.** *Let $H$ be a real Hilbert space and $K : E_X \times E_X \to \mathcal{L}(H)$ be an operator valued positive-semidefinite (psd) kernel such that $K(x,x') = K(x',x)^*$ for all $x,x' \in E_X$, and for all $x_1,\ldots,x_n \in E_X$ and $h_i, h_j \in H$,*

$$\sum_{i,j=1}^{n} \langle h_i, K(x_i, x_j) h_j \rangle_H \geq 0.$$

Fix $K$, $x \in E_X$, and $h \in H$, $[K_x h](\cdot) := K(\cdot, x)h$ defines a function from $E_X$ to $H$. We now consider

$$\mathcal{G}_{\mathrm{pre}} := \mathrm{span} \{K_x h \mid x \in E_X, h \in H\}$$

with inner product on $\mathcal{G}_{\mathrm{pre}}$ by linearly extending the expression

$$\langle K_x h, K_{x'} h' \rangle_{\mathcal{G}} := \langle h, K(x,x') h' \rangle_H. \tag{1}$$

Let $\mathcal{G}$ be the completion of $\mathcal{G}_{\mathrm{pre}}$ with respect to this inner product. We call $\mathcal{G}$ the vRKHS induced by the kernel $K$. The space $\mathcal{G}$ is a Hilbert space consisting of functions from $E_X$ to $H$ with the reproducing property

$$\langle F(x), h \rangle_H = \langle F, K_x h \rangle_{\mathcal{G}}, \tag{2}$$

for all $F \in \mathcal{G}, h \in H$ and $x \in E_X$. For all $F \in \mathcal{G}$ we obtain

$$\|F(x)\|_H \leq \|K(x,x)\|^{1/2} \|F\|_{\mathcal{G}}, \quad x \in E_X.$$

Since the inner product given by Eq. (1) implies that $K_x$ is a bounded operator for all $x \in E_X$. For all $F \in \mathcal{G}$ and $x \in E_X$, Eq. (2) can be written as $F(x) = K_x^* F$. The linear operators $K_x : H \to \mathcal{G}$ and $K_x^* : \mathcal{G} \to H$ are bounded with

$$\|K_x\| = \|K_x^*\| = \|K(x,x)\|^{1/2}$$

and we have $K_x^* K_{x'} = K(x,x'), x, x' \in E_X$. In the following, we will denote $\mathcal{G}$ as the vRKHS induced by the kernel $K : E_X \times E_X \to \mathcal{L}(\mathcal{H}_Y)$ with

$$K(x,x') := k_X(x,x')\mathrm{Id}_{\mathcal{H}_Y}, x, x' \in E_X.$$

An important property of $\mathcal{G}$ is that elements in $\mathcal{G}$ are isometric to Hilbert-Schmidt operators between $\mathcal{H}_X$ and $\mathcal{H}_Y$.

**Theorem 1** (Theorem 4.4 in [26]). *Let $\mathcal{H}_X$ and $\mathcal{H}_Y$ be real-valued RKHS with kernel $k_X$ and $k_Y$ respectively. For $f_Y \in \mathcal{H}_Y$ and $g_X \in \mathcal{H}_X$, define the map $\bar{\Psi}$ on the elementary tensors as*

$$\left[ \bar{\Psi} \left( f_Y \otimes g_X \right) \right] (x) := g_X(x) f_Y = (f_Y \otimes g_X) \phi_X(x).$$

*We then have that $\bar{\Psi}$ defines an isometric isomorphism between $S_2(\mathcal{H}_X, \mathcal{H}_Y)$ and $\mathcal{G}$ through linearity and completion.*

More details regarding Theorem 1 can be found in [26, Theorem 4.4]. The isometric isomorphism $\bar{\Psi}$ induces the operator reproducing property stated below.

**Corollary 1.** *For every function $F \in \mathcal{G}$ there exists an operator $C := \bar{\Psi}^{-1}(F) \in S_2(\mathcal{H}_X, \mathcal{H}_Y)$ such that*

$$F(x) = C\phi_X(x) \in \mathcal{H}_Y,$$

*for all $x \in E_X$ with $\|C\|_{S_2(\mathcal{H}_X, \mathcal{H}_Y)} = \|F\|_{\mathcal{G}}$ and vice versa. Conversely, for any pair $F \in \mathcal{G}$ and $C \in S_2(\mathcal{H}_X, \mathcal{H}_Y)$, we have $C = \bar{\Psi}^{-1}(F)$ as long as $F(x) = C\phi_X(x)$.*

The proof of Corollary 1 is a simple extension of Lemma 15 in [7] and Corollary 4.5 in [26]. Corollary 1 shows that the vRKHS $\mathcal{G}$ is generated via the space of Hilbert-Schmidt operators $S_2(\mathcal{H}_X, \mathcal{H}_Y)$

$$\mathcal{G} = \{F : E_X \to \mathcal{H}_Y | F = C\phi_X(\cdot), C \in S_2(\mathcal{H}_X, \mathcal{H}_Y)\}.$$

**Conditional Mean Embedding:** A particular advantage of kernel methods is its convenience of operating probability distributions, see [27, 30] for examples. This is through the so called kernel mean embedding [2, 34, 17]. Assuming the integrability condition $\int_{E_X} \sqrt{k_X(x,x)} d\pi(x) < \infty$ (which is satisfied when the kernel is almost surely bounded), we define the kernel mean embedding $\mu_X(\cdot) = \int_{E_X} k_X(\cdot, x) d\pi(x)$. It is easy to show that for each $f \in \mathcal{H}_X$, $\int_{E_X} f(x) d\pi(x) = \langle f, \mu_X \rangle_{\mathcal{H}_X}$. Replacing $\pi$ with the conditional distribution, we obtain the kernel conditional mean embedding as defined in [29, 18].

**Definition 2.** *The $\mathcal{H}_Y$-valued conditional mean embedding (CME) for the Markov kernel $p(x, dy)$ is defined as*

$$F_*(x) := \int_{E_Y} \phi_Y(y) p(x, dy) = \mathbb{E}\left[\phi_Y(Y) | X = x\right] \in L_2(E_X, \mathcal{F}_{E_X}, \pi; \mathcal{H}_Y) \qquad (3)$$

By the reproducing property, we have $\mathbb{E}[f_Y(Y) | X = x] = \langle f_Y, F_*(x) \rangle_{\mathcal{H}_Y}, \forall f_Y \in \mathcal{H}_Y$ and $x \in E_X$. The approximation of $F_*$ is a key concept in kernel methods. By [26], suppose we impose Assumptions 1-3 together with two additional assumptions: i) $\mathcal{H}_X \subseteq C_0(E_X)$ where $C_0(E_X)$ is the space of continuous functions vanishing at infinity[2]; and ii) $\mathcal{H}_X$ is dense in $L_2(\pi)$, then we have that $\mathcal{G}$ is dense in $L_2(E_X, \mathcal{F}_{E_X}, \pi; \mathcal{H}_Y)$. As a result, for any $\delta > 0$, there is an $F \in \mathcal{G}$ such that $\|F - F_*\|_{L_2} < \delta$. Hence, in the literature, we often assume the so-called *well-specified case* to obtain a closed-form solution,

$$F_* \in \mathcal{G}. \qquad (4)$$

It is shown in [18, Theorem 5.3] and [26, Corollary 5.6 and Remark 5.8] that $F_*$ admits a closed form expression under Eq. (4) via

$$F_*(x) = (C_{XX}^\dagger C_{XY})^* \phi_X(x),$$

where $C_{YX} = \mathbb{E}[\phi_Y(Y) \otimes \phi_X(X)]$ and $C^\dagger$ denotes the pseudoinverse of $C$.

**Remark 2.** *We point out that in the original derivations, the CME is written as $F_*(x) = C_{YX} C_{XX}^\dagger \phi_X(x)$ [37, 12, 13]. However, $C_{XX}^\dagger$ is not globally defined if $\mathcal{H}_X$ is infinite-dimensional. Hence the expression $C_{YX} C_{XX}^\dagger \phi_X(x)$ is problematic, as we expect $F_*$ to be defined for all $x \in E_X$ based on the Markov kernel $p$. In the well-specified scenario, [18] corrected this issue by defining the CME as $(C_{XX}^\dagger C_{XY})^* \phi_X(x)$. It is shown that in this case, $(C_{XX}^\dagger C_{XY})^*$ is bounded (actually Hilbert-Schmidt, see also [19]), and hence globally defined. The connection of this corrected operator-theoretic perspective to the well-specified regression scenario was established in [26].*

---

[2]This is satisfied if $k_X$ is bounded and $k_X(\cdot, x) \in C_0(E_X)$ for $\pi$-almost all $x \in E_X$.

Once we have the closed-form solution, a natural question to ask is how to estimate the CME. Indeed, this has been extensively studied in [15, 29, 41]. Given a data set $D = \{(x_i, y_i)\}_{i=1}^n$ independently and identically sampled from the joint distribution of $X$ and $Y$, a regularized estimate of $F_*$ is the solution of the following optimization problem:

$$\hat{C}_{Y|X,\lambda} := \underset{C \in S_2(\mathcal{H}_X, \mathcal{H}_Y)}{\arg\min} \frac{1}{n} \sum_{i=1}^n \|\phi_Y(y_i) - C\phi_X(x_i)\|_{\mathcal{H}_Y}^2 + \lambda\|C\|_{S_2(\mathcal{H}_X, \mathcal{H}_Y)}^2, \tag{5}$$

$\hat{F}_\lambda(\cdot) := \bar{\Psi}\left(\hat{C}_{Y|X,\lambda}\right)(\cdot) = \hat{C}_{Y|X,\lambda}\phi_X(\cdot)$, where $\lambda$ is the regularization parameter. Implicit in the construction, however, is the assumption $F_* \in \mathcal{G}$ that the solution is well-specified. We provide a few remarks regarding this assumption:

**Remark 3.** *i) In the literature, the prevalent definition of well-specifiedness is through*

$$\mathbb{E}[f_Y(Y)|X = \cdot] \in \mathcal{H}_X, \forall f_Y \in \mathcal{H}_Y, \tag{6}$$

*see e.g. [12, 37, 13] for details. However, this definition is not equivalent to that in Eq. (4). Specifically, assuming $F_* \in \mathcal{G}$ implies that Eq. (6) holds. Nonetheless, the reverse is not true. In particular, there exist concrete examples satisfying Eq. (6), but the corresponding operator representative of the CME is not Hilbert–Schmidt (see Section D in Appendix for details). To avoid confusion, we refer to Eq. (4) as the well-specified case hereafter.*

*ii) The conventional assumption Eq. (6) can actually be refined via the inclusion map $I_\pi$. In particular, since $I_\pi$ is an inclusion map from $\mathcal{H}_X$ to $L_2(\pi)$ and $\mathbb{E}[f_Y(Y)|X = \cdot] \in \mathcal{H}_X$, we can apply $I_\pi$ to $\mathbb{E}[f_Y(Y)|X = \cdot]$. In addition, we are only interested in the case where $I_\pi(\mathbb{E}[f_Y(Y)|X = \cdot]) \neq 0$. As a result, the refined definition should be*

$$\mathbb{E}[f_Y(Y)|X = \cdot] \in (\ker I_\pi)^\perp, \forall f_Y \in \mathcal{H}_Y. \tag{7}$$

We now characterize the Hilbert spaces used to define the CME in the misspecified setting.

**Real-valued Interpolation Space:** We review the results of [40, 11] that set out the eigendecompositions of $L_X$ and $C_{XX}$, and apply these in constructing the interpolation spaces used for the misspecified setting. By the spectral theorem for self-adjoint compact operators, there exists an at most countable index set $I$, a non-increasing sequence $(\mu_i)_{i \in I} > 0$, and a family $(e_i)_{i \in I} \in \mathcal{H}_X$, such that $([e_i])_{i \in I}$ is an orthonormal basis (ONB) of $\overline{\operatorname{ran} I_\pi} \subseteq L_2(\pi)$ and $(\mu_i^{1/2} e_i)_{i \in I}$ is an ONB of $(\ker I_\pi)^\perp \subseteq \mathcal{H}_X$, and we have

$$L_X = \sum_{i \in I} \mu_i \langle \cdot, [e_i]\rangle_{L_2(\pi)}[e_i], \qquad C_{XX} = \sum_{i \in I} \mu_i \langle \cdot, \mu_i^{\frac{1}{2}} e_i\rangle_{\mathcal{H}_X} \mu_i^{\frac{1}{2}} e_i.$$

For $\alpha \geq 0$, we define the $\alpha$-interpolation space [40] by

$$[\mathcal{H}]_X^\alpha := \left\{ \sum_{i \in I} a_i \mu_i^{\alpha/2}[e_i] : (a_i)_{i \in I} \in \ell_2(I) \right\} \subseteq L_2(\pi),$$

equipped with the $\alpha$-power norm

$$\left\| \sum_{i \in I} a_i \mu_i^{\alpha/2}[e_i] \right\|_{[\mathcal{H}]_X^\alpha} := \|(a_i)_{i \in I}\|_{\ell_2(I)} = \left( \sum_{i \in I} a_i^2 \right)^{1/2}.$$

More broadly, we sometimes need to deal with function of the form $f + c$ where $f \in [\mathcal{H}]_X^\alpha$ and $c \in \mathbb{R}$. For this, we follow the classical definition of the direct sum of two Hilbert spaces [8] and define the $\alpha$-power norm for $f + c$ as

$$\|f + c\|_{[\mathcal{H}]_X^\alpha} = \|c\|_{\mathbb{R}} + \|f\|_{[\mathcal{H}]_X^\alpha}. \tag{8}$$

For $(a_i)_{i \in I} \in \ell_2(I)$, the $\alpha$-interpolation space becomes a Hilbert space with inner product defined as

$$\left\langle \sum_{i \in I} a_i(\mu_i^{\alpha/2}[e_i]), \sum_{i \in I} b_i(\mu_i^{\alpha/2}[e_i]) \right\rangle_{[\mathcal{H}]_X^\alpha} = \sum_{i \in I} a_i b_i.$$

$$S_2(L_2(\pi), \mathcal{H}_Y) \xrightarrow[\Psi]{} L_2(\pi; \mathcal{H}_Y)$$

$$\mathcal{I}_\pi \Big\uparrow$$

$$S_2(\mathcal{H}_X, \mathcal{H}_Y) \xrightarrow[\bar{\Psi}]{} \mathcal{G}$$

Figure 1: $\Psi$ and $\bar{\Psi}$ are the bijective linear operators that define the respective isomorphisms between each pair of spaces. The precise form of the isomorphisms is given in the appendix. $\mathcal{I}_\pi$ denotes the canonical embedding between the two Hilbert-Schmidt spaces.

Moreover, $\left( \mu_i^{\alpha/2} [e_i] \right)_{i \in I}$ forms an ONB of $[\mathcal{H}]_X^\alpha$ and consequently $[\mathcal{H}]_X^\alpha$ is a separable Hilbert space. In the following, we use the abbreviation $\| \cdot \|_\alpha := \| \cdot \|_{[\mathcal{H}]_X^\alpha}$. For $\alpha = 0$ we have $[\mathcal{H}]_X^0 = \overline{\operatorname{ran} I_\pi} \subseteq L_2(\pi)$ with $\| \cdot \|_0 = \| \cdot \|_{L_2(\pi)}$. Moreover, for $\alpha = 1$ we have $[\mathcal{H}]_X^1 = \operatorname{ran} I_\pi$ and $[\mathcal{H}]_X^1$ is isometrically isomorphic to the closed subspace $(\ker I_\pi)^\perp$ of $\mathcal{H}_X$ via $I_\pi$, i.e. $\|[f]\|_1 = \|f\|_{\mathcal{H}_X}$ for $f \in (\ker I_\pi)^\perp$. For $0 < \beta < \alpha$, we have

$$[\mathcal{H}]_X^\alpha \hookrightarrow [\mathcal{H}]_X^\beta \hookrightarrow [\mathcal{H}]_X^0 \subseteq L_2(\pi). \tag{9}$$

Moreover, under Assumptions 1-3, if we further assume that $supp(\pi) = E_X$ and $\mathcal{H}_X$ is dense in $L_2(\pi)$, we have $[\mathcal{H}]_X^0 = L_2(\pi)$.

## 3  Approximation of CME with Vector-valued Interpolation Space

In this section, we deal with the misspecified setting where $F_* \notin \mathcal{G}$. To do this, we first define the *vector-valued interpolation space* via the tensor product space. We now recall from Remark 1 that $L_2(\pi; \mathcal{H}_Y)$ is isomorphic to $S_2(L_2(\pi), \mathcal{H}_Y)$ and we denote by $\Psi$ the isomorphism between the two spaces. Similarly, we have $\mathcal{G} \simeq S_2(\mathcal{H}_X, \mathcal{H}_Y)$ and we denote by $\bar{\Psi}$ the isomorphism between both spaces in accordance with Theorem 1. This is summarized in Figure 1. The second chain of spaces is not isometric to the first but can be naturally embedded into the first as follows. Recall that we denote by $I_\pi : \mathcal{H}_X \to L_2(\pi)$ the embedding that maps each function to its equivalent class, $I_\pi(f) = [f]$. We therefore naturally define the embedding $\mathcal{I}_\pi : S_2(\mathcal{H}_X, \mathcal{H}_Y) \to S_2(L_2(\pi), \mathcal{H}_Y)$ through $\mathcal{I}_\pi(g \otimes f) = g \otimes I_\pi(f) = g \otimes [f]$ for all $f \in \mathcal{H}_X, g \in \mathcal{H}_Y$, and obtain the extension to the whole space by linearity and continuity. Therefore, for $F \in \mathcal{G}$ we define $[F] := \Psi \circ \mathcal{I}_\pi \circ \bar{\Psi}^{-1}(F)$. In the rest of the paper, every embedding will be denoted using the notation $[ \cdot ]$. Strict notation would require us to write $[ \cdot ]_\pi$ due to dependence on the measure $\pi$, but we omit the subscript for ease of notation.

**Definition 3.** *Suppose that we are given real-valued kernels $k_X$ and $k_Y$ with associated RKHS $\mathcal{H}_X$ and $\mathcal{H}_Y$ and let $[\mathcal{H}]_X^\alpha$ be the real-valued interpolation space associated to $\mathcal{H}_X$ with some $\alpha \geq 0$. Since $[\mathcal{H}]_X^\alpha \subseteq L_2(\pi)$, it is natural to define the vector-valued interpolation space $[\mathcal{G}]^\alpha$ as*

$$[\mathcal{G}]^\alpha := \Psi\left(S_2([\mathcal{H}]_X^\alpha, \mathcal{H}_Y)\right) = \{F \mid F = \Psi(C), \ C \in S_2([\mathcal{H}]_X^\alpha, \mathcal{H}_Y)\}.$$

*$[\mathcal{G}]^\alpha$ is a Hilbert space equipped with the norm*

$$\|F\|_\alpha := \|C\|_{S_2([\mathcal{H}]_X^\alpha, \mathcal{H}_Y)} \qquad (F \in [\mathcal{G}]^\alpha),$$

*where $C = \Psi^{-1}(F)$. For $\alpha = 0$, we retrieve,*

$$\|F\|_0 = \|C\|_{S_2(L_2(\pi), \mathcal{H}_Y)}.$$

**Remark 4.** *The vector-valued interpolation space $[\mathcal{G}]^\alpha$ allows us to study the CME in the misspecified case. To see this, we note that by Eq. (3), we have $F_* \in L_2(E_X, \mathcal{F}_{E_X}, \pi; \mathcal{H}_Y)$. In light of Eq. (9), for $0 < \beta < \alpha$ we have*

$$[\mathcal{G}]^\alpha \hookrightarrow [\mathcal{G}]^\beta \hookrightarrow [\mathcal{G}]^0 \subseteq L_2(E_X, \mathcal{F}_{E_X}, \pi; \mathcal{H}_Y).$$

*Again, once we further assume that $supp(\pi) = E_X$ and $\mathcal{H}_X$ is dense in $L_2(\pi)$, we have $[\mathcal{G}]^0 = L_2(E_X, \mathcal{F}_{E_X}, \pi; \mathcal{H}_Y)$. Hence, while the well-specified case corresponds to $F_* \in \mathcal{G}$, the misspecified case amounts to assuming that $F_* \in [\mathcal{G}]^\alpha$ for some $0 < \alpha < 1$, and relaxes the well-specified assumption in Eq. (4).*

# 4 Learning Rate for CME

In this section, we derive the learning rate for the difference between $[\hat{F}_\lambda]$ and $F_*$ in the interpolation norm. We first state additional assumptions that are needed in our derivations. As our assumptions match those of [11], we include the corresponding labels from [11] for ease of reference.

5. Recall that $(\mu_i)_{i \in I}$ are the eigenvalues of $C_{XX}$. For some constants $c_2 > 0$ and $p \in (0, 1]$ and for all $i \in I$,

$$\mu_i \leq c_2 i^{-1/p} \tag{EVD}$$

6. For $\alpha \in (p, 1]$, the inclusion map $I_\pi^{\alpha,\infty} : [\mathcal{H}]_X^\alpha \hookrightarrow L_\infty(\pi)$ is continuous, and there is a constant $A > 0$ such that

$$\|I_\pi^{\alpha,\infty}\|_{[\mathcal{H}]_X^\alpha \to L_\infty(\pi)} \leq A \tag{EMB}$$

7. There exists $0 < \beta \leq 2$ such that

$$F_* \in [\mathcal{G}]^\beta \tag{SRC}$$

We let $C_{Y|X} := \Psi^{-1}(F_*) \in S_2([\mathcal{H}]_X^\beta, \mathcal{H}_Y)$ and we call $C_{Y|X}$ the conditional mean embedding operator.

(EVD) is a standard assumption on the eigenvalue decay of the integral operator (see more details in [4, 11, 41]). (EMB) is referred as the embedding property in [11] and it can be shown that it implies $\sum_{i \in I} \mu_i^\alpha e_i^2(x) \leq A^2$ for $\pi$-almost all $x \in E_X$ ([11] Theorem 9). Since we assume $k_X$ to be bounded, the embedding property always hold true when $\alpha = 1$. Furthermore, (EMB) implies a polynomial eigenvalue decay of order $1/\alpha$, which is why we take $\alpha \geq p$. (SRC) is justified by Remark 4 and is often referred as the source condition in literature ([4, 11, 23, 24]). It imposes the smoothness assumption on the target CME operator $F_*$. In particular, when $\beta \geq 1$, the source condition implies that $F_*$ has a representative from $\mathcal{G}$, indicating the well specified scenario. However, once we let $\beta < 1$, we are in the misspecified learning setting, which is the main interest in this manuscript. Finally, in computing the learning rate for real-valued regression, one often needs the so-called (MOM) condition on the Markov kernel $p(x, dy)$ (see [4, 11, 41] for more details). The generalization to our setting would amount to assume that there exists constants $\sigma, R > 0$ such that

$$\mathbb{E}\left[\|\phi_Y(Y) - F_*(x)\|_{\mathcal{H}_Y}^q \mid X = x\right] \leq \frac{1}{2}q!\sigma^2 R^{q-2},$$

for $\pi$-almost surely all $x \in E_X$ and all $q \geq 2$. The reason for requiring (MOM) in the scalar regression setting is that we do not usually have $|Y| < \infty$ almost surely. In our setting, however, Assumption 3 implies $\|\phi_Y(y) - F_*(x)\|_{\mathcal{H}_Y} \leq 2\kappa_Y$ for $\pi$-almost all $x \in E_X$ and $\nu$-almost all $y \in E_Y$. Therefore, (MOM) is automatically satisfied with $\sigma = R = 2\kappa_Y$.

**Remark 5.** *We remark that in [41], a variant of SRC condition is employed. In particular, instead of assuming $F_* \in [\mathcal{G}]^\beta$, they impose the assumption that $F_* \in \mathcal{G}^\beta \simeq S_2(\mathcal{H}_X^\beta, \mathcal{H}_Y)$, where $\mathcal{H}_X^\beta$ is an RKHS with corresponding kernel defined as $k_X^\beta(\cdot, x) = \sum_i \mu_i^\beta e_i(\cdot)e_i(x)$. Comparing to $[\mathcal{G}]^\beta \simeq S_2([\mathcal{H}]_X^\beta, \mathcal{H}_Y)$, there are two shortcomings that arise when working with $\mathcal{G}^\beta$.*

*First, $\mathcal{H}_X^\beta$ denotes the interpolating RKHS consisting of continuous functions only, while $[\mathcal{H}]_X^\beta$ is the interpolating Hilbert space, where elements are defined through an equivalence classes. Hence, by working with $\mathcal{G}^\beta$, the implicit assumption is that $F_*$ is a continuous function. On the other hand, assuming $F_* \in [\mathcal{G}]^\beta$ avoids the continuity requirement. In particular, we have $\mathcal{H}_X^\beta \subseteq [\mathcal{H}]_X^\beta$ for any $\beta > 0$. Therefore, our (SRC) condition applies to a more general setting.*

*Second, and more importantly, the construction of $\mathcal{H}_X^\beta$ relies on the condition that $\sum_i \mu_i^\beta e_i^2(x) < \infty$, as pointed out in [40]. Failing this, the kernel $k_X^\beta$ associated with the interpolating RKHS $\mathcal{H}_X^\beta$ is unbounded, indicating $\mathcal{H}_X^\beta$ is not well-defined. Under (EVD), this effectively amounts to require that $\beta \geq p$. When kernel has slow eigenvalue decay (as for the Matérn kernel), $p$ can be close to $1$. Results obtained using $\mathcal{G}^\beta$ while requiring $\beta > p$ are very close to the well-specified case. By contrast, $[\mathcal{H}]_X^\beta$ is always well-defined as a subspace of $L_2(\pi)$ for any $\beta > 0$, even if $\sum_i \mu_i^\beta e_i^2(x) < \infty$ does not hold.*

**Remark 6.** *It is important to assume $\beta > 0$ in (SRC), as our results do not apply when $\beta = 0$. The $\beta = 0$ setting arises for instance when $Y \perp X$, or when both $Y = X$ and $\mathcal{H}_Y = \mathcal{H}_X$. In the former*

*case, it is easy to see that the CME is the constant function (w.r.t $X$) $F_*(x) = \mu_Y = \int_{E_Y} \phi_Y(y)d\nu(y)$. In the latter case, $F_*(x) = \phi_X(x)$, and the CME is the identity operator. These functions are not covered by $[\mathcal{G}]^\beta$ for any $\beta > 0$ (see Appendix D for details).*

We now provide an upper bound on the learning rate.

**Theorem 2.** *Let Assumptions 1-3, (EVD), (EMB) and (SRC) with $0 < \beta \leq 2$ hold, and let $0 \leq \gamma \leq 1$ with $\gamma < \beta$,*

1. *In the case $\beta + p \leq \alpha$ and $\lambda_n = \Theta\left((n/\log^r(n))^{-\frac{1}{\alpha}}\right)$, for some $r > 1$, there is a constant $K > 0$ independent of $n \geq 1$ and $\tau \geq 1$ such that*

$$\left\|[\hat{F}_\lambda] - F_*\right\|_\gamma^2 \leq \tau^2 K \left(\frac{n}{\log^r n}\right)^{-\frac{\beta-\gamma}{\alpha}}$$

   *is satisfied for sufficiently large $n \geq 1$ with $P^n$-probability not less than $1 - 4e^{-\tau}$.*

2. *In the case $\beta + p > \alpha$ and $\lambda_n = \Theta\left(n^{-\frac{1}{\beta+p}}\right)$, there is a constant $K > 0$ independent of $n \geq 1$ and $\tau \geq 1$ such that*

$$\left\|[\hat{F}_\lambda] - F_*\right\|_\gamma^2 \leq \tau^2 K n^{-\frac{\beta-\gamma}{\beta+p}}$$

   *is satisfied for sufficiently large $n \geq 1$ with $P^n$-probability not less than $1 - 4e^{-\tau}$.*

Theorem 2 provides the finite sample $\gamma$-norm learning rate for the empirical CME estimator defined in Eq. (5). It states that the learning rate for $[\hat{F}_\lambda]$ is governed by the interplay between $p$, $\alpha$, and $\beta$. Intuitively, $p$ describes the decay rate of the eigenvalues $(\mu_i)_{i \in I}$, $\alpha$ determines the boundedness of the interpolation kernel (and has maximum value of 1 according to our assumption), $\beta$ characterizes the smoothness of the target CME operator.

To simplify the discussion, we may focus on the $L_2(E_X, \mathcal{F}_{E_X}, \pi; \mathcal{H}_Y)$ learning rate, corresponding to $\gamma = 0$. The exponent $\beta/\max\{\alpha, \beta + p\}$ explicitly provides the learning rate for the CME operator. For example, if we have $\alpha \leq \beta$, we obtain a learning rate of $\beta/(\beta + p)$. In particular, for a Gaussian kernel, $p$ and $\alpha$ are arbitrarily close to 0, and our learning rate can achieve a fast $O(1/n)$ rate up to a logarithmic factor. If a kernel with slow eigenvalue decay is used, such as the Matérn kernel, we can obtain the minimax optimal learning rate $n^{-1/2}$ up to logarithmic factors if we have $p \leq \beta$. Finally, in the worst case where $\beta$ is close to 0, the learning rate can be arbitrarily slow.

## 5 Lower Bound

Our final theorem provides a lower bound for the convergence rate, which allows us to confirm the optimality of our learning rate. In deriving the lower bound, we need an extra assumption

8. For some constants $c_1, c_2 > 0$ and $p \in (0, 1]$ and for all $i \in I$,

$$c_1 i^{-1/p} \leq \mu_i \leq c_2 i^{-1/p} \qquad \text{(EVD+)}$$

**Theorem 3.** *Let $k_X$ be a kernel on $E_X$ such that Assumptions 1-3 hold and $\pi$ be a probability distribution on $E_X$ such that (EVD+) and (EMB) hold $0 < p \leq \alpha \leq 1$. Then for all $0 < \beta \leq 2$, $0 \leq \gamma \leq 1$ with $\gamma < \beta$ there exist constants $K_0, K, s > 0$ such that for all learning methods $D \to \hat{F}_D$ ($D := \{(x_i, y_i)\}_{i=1}^n$), all $\tau > 0$, and all sufficiently large $n \geq 1$ there is a distribution $P$ defined on $E_X \times E_Y$ used to sample $D$, with marginal distribution $\pi$ on $E_X$, such that (SRC) with respect to $\beta$ is satisfied, and with $P^n$-probability not less than $1 - K_0\tau^{1/s}$,*

$$\|[\hat{F}_D] - F_*\|_\gamma^2 \geq \tau^2 K n^{-\frac{\max\{\alpha,\beta\}-\gamma}{\max\{\alpha,\beta\}+p}}.$$

Theorem 3 states that under the assumtions of Theorem 2 and (EVD+), no learning method can achieve a learning rate faster than $n^{-\frac{\max\{\alpha,\beta\}}{\max\{\alpha,\beta\}+p}}$ in $L_2$ norm. To our knowledge, this is the first analysis that demonstrates the lower rate for CME learning. In the context of regularized regression,

[4, 39, 3] provide a similar lower bound on the learning rate. However, a key difference in our analysis is that the output of the regression learning now lives in an infinite dimensional RKHS $\mathcal{H}_Y$, rather than in $\mathbb{R}$. Our analysis reveals that in the case where $\alpha \leq \beta$, the obtained upper rate in Theorem 2 is optimal, i.e., $O(n^{-\frac{\beta}{\beta+p}})$. In particular, when $k_X$ is an exponentiated quadratic kernel on a compact set $E_X \subset \mathbb{R}^d$ with Lipschitz boundary, (EMB) is satisfied with any $\alpha \in (0,1)$ [17, see Corollary 4.13]. As a result, the optimal rate is attained as long as $\beta > 0$. We point out that finding the optimal rate for $\beta < \alpha$ remains a challenge, and is an open problem when the output is $\mathbb{R}$.

**Remark 7.** *Theorem 3 states that the upper bound and the lower bound match when $\beta > \alpha$. In particular, for exponential kernels such as Gaussian and Laplacian kernels with subgaussian distributions, the eigenvalues for the covariance operator have geometric decay rate. In these cases, $\alpha$ is arbitrarily close to $0$. Hence, as long as $\beta > 0$, we will have $\alpha < \beta$. In other words, for commonly used kernels, CME learning can always obtain the optimal fast rate $n^{-\frac{\beta}{\beta+p}}$.*

## 6 Conclusion

In this paper, we provide a rigorous theoretical foundation for approximating the CME operator, and study the statistical learning rate. Utilizing recently developed interpolation space techniques, we first define the vector-valued interpolation space $[\mathcal{G}]^\alpha$. This allows to define the target CME operator in the larger interpolation space $[\mathcal{G}]^\alpha$, in contrast to the well-specified setting where $F_* \in \mathcal{G}$. By doing so, we are able to study the convergence rate of the empirical CME operator in the misspecified scenario. We then provide a $\gamma$-norm learning rate for the CME without any assumption on the interplay between $\beta$ and $p$, with matching lower bound. Our analysis shows that under appropriate conditions, we can obtain a fast $O(\log n/n)$ convergence rate, which matches the rate obtained in the existing literature for finite dimensional $\mathcal{H}_Y$. In more challenging settings, we still obtain the minimax optimal rate $O(n^{-1/2})$.

Looking beyond the present work, our current interpolation space setting indicates that the convergence rate can be arbitrarily slow if $\beta \to 0$. This prevents learning the constant function, which plays a crucial role in completing the theory of the CME, as pointed out by [18]. Addressing this challenge is an important direction of future research.

**Acknowledgement:** The authors wish to thank Peter Orbanz and Bharath Sriperumbudur for fruitful discussions and proofreading. Zhu Li, Dimitri Meunier, and Arthur Gretton were supported by the Gatsby Charitable Foundation. Mattes Mollenhauer was supported by the Deutsche Forschungsgemeinschaft (DFG) through grant EXC 2046 "MATH+"; Project Number 390685689, Project IN-8 "Infinite-Dimensional Supervised Least Squares Learning as a Noncompact Regularized Inverse Problem".

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
