# A    Proof of Theorem 2

**Sketch of Proof.**    Recall that $\hat{F}_\lambda \in \mathcal{G}$ is defined as $\hat{F}_\lambda := \bar{\Psi}\left(\hat{C}_{Y|X,\lambda}\right)$ where $\hat{C}_{Y|X,\lambda}$ is solution of Eq. (5). We introduce the theoretical estimator that solves the regression problem in population,

$$C_{Y|X,\lambda} := \underset{C \in S_2(\mathcal{H}_X, \mathcal{H}_Y)}{\arg\min} \mathbb{E}_{XY} \|\phi_Y(Y) - C\phi_X(X)\|^2_{\mathcal{H}_Y} + \lambda \|C\|^2_{S_2(\mathcal{H}_X, \mathcal{H}_Y)}$$

$$F_\lambda := \bar{\Psi}\left(C_{Y|X,\lambda}\right) \tag{10}$$

It can be readily shown (see for example [26]) that

$$C_{Y|X,\lambda} = C_{YX}\left(C_{XX} + \lambda Id_{\mathcal{H}_X}\right)^{-1}$$

$$\hat{C}_{Y|X,\lambda} = \hat{C}_{YX}\left(\hat{C}_{XX} + \lambda Id_{\mathcal{H}_X}\right)^{-1},$$

where $Id_{\mathcal{H}_X}$ is the identity operator and

$$C_{XX} = \mathbb{E}[\phi_X(X) \otimes \phi_X(X)] \qquad C_{YX} = \mathbb{E}[\phi_Y(Y) \otimes \phi_X(X)]$$

$$\hat{C}_{XX} = \frac{1}{n}\sum_{i=1}^{n} \phi_X(x_i) \otimes \phi_X(x_i) \qquad \hat{C}_{YX} = \frac{1}{n}\sum_{i=1}^{n} \phi_Y(y_i) \otimes \phi_X(x_i). \tag{11}$$

Finally, recall that the CME $F_*$ is in $L_2(E_X, \mathcal{F}_{E_X}, \pi; \mathcal{H}_Y)$ and the CME *operator* is defined as $C_{Y|X} := \Psi^{-1}(F_*)$. From the definition of the vector-valued interpolation norm we introduce the following decomposition,

$$\left\|[\hat{F}_\lambda] - F_*\right\|_\gamma \leq \left\|\left[\hat{F}_\lambda - F_\lambda\right]\right\|_\gamma + \|[F_\lambda] - F_*\|_\gamma \tag{12}$$

$$= \left\|\left[\hat{C}_{Y|X,\lambda} - C_{Y|X,\lambda}\right]\right\|_{S_2\left([\mathcal{H}]_X^\gamma, \mathcal{H}_Y\right)} + \left\|[C_{Y|X,\lambda}] - C_{Y|X}\right\|_{S_2\left([\mathcal{H}]_X^\gamma, \mathcal{H}_Y\right)} \tag{13}$$

We can see that the error for the first term is mainly due to the sample approximation. We therefore refer to the first term as the *Variance*. We refer to the second term as the *Bias*. Our proof of convergence of the bias adapts the proof in [31, Theorem 6] and [11], and utilizes the fact that $C_{Y|X}$ is Hilbert-Schmidt to obtain a sharp rate.

## A.1    Bounding the Bias

In this section, we establish the bound on the bias. The key insight is that thanks to [1, Theorem 12.6.1], the conditional mean embedding can be expressed as a Hilbert-Schmidt operator in the misspecified case. We then exploit the proof techniques from the bias consistency result of [31, Theorem 6] and [11].

**Lemma 1.** *If $F_* \in [\mathcal{G}]^\beta$ is satisfied for some $0 \leq \beta \leq 2$, then the following bound is satisfied, for all $\lambda > 0$ and $0 \leq \gamma \leq \beta$:*

$$\|[F_\lambda] - F_*\|^2_\gamma \leq \|F_*\|^2_\beta \lambda^{\beta - \gamma} \tag{14}$$

*Proof.* We first recall that since $F_* \in [\mathcal{G}]^\beta$, $F_* = \Psi\left(C_{Y|X}\right)$ with $C_{Y|X} \in S_2([\mathcal{H}]_X^\beta, \mathcal{H}_Y)$, furthermore $F_\lambda = \bar{\Psi}\left(C_{Y|X,\lambda}\right)$ with $C_{Y|X,\lambda} \in S_2(\mathcal{H}_X, \mathcal{H}_Y)$. Hence, $\|[F_\lambda] - F_*\|_\gamma = \left\|[C_{Y|X,\lambda}] - C_{Y|X}\right\|_{S_2\left([\mathcal{H}]_X^\gamma, \mathcal{H}_Y\right)}$ and $\|F_*\|_\beta = \left\|C_{Y|X}\right\|_{S_2\left([\mathcal{H}]_X^\beta, \mathcal{H}_Y\right)}$. We first decompose $[C_{Y|X,\lambda}] - C_{Y|X}$, followed by computing the upper bound of the bias. Since $C_{Y|X} \in S_2([\mathcal{H}]_X^\beta, \mathcal{H}_Y) \subseteq S_2(\overline{\operatorname{ran} I_\pi}, \mathcal{H}_Y)$, $C_{Y|X}$ admits the decomposition

$$C_{Y|X} = \sum_{i \in I}\sum_{j \in J} a_{ij} d_j \otimes [e_i].$$

where $(d_j)_{j \in J}$ is any basis of $\mathcal{H}_Y$. On the other hand, $C_{Y|X,\lambda} = C_{YX}\left(C_{XX} + \lambda Id_{\mathcal{H}_X}\right)^{-1}$. Since $\left(\mu_i^{1/2}e_i\right)_{i \in I}$ is an ONB of $(\ker I_\pi)^\perp$, we can complete it with an at most countable basis $(\bar{e}_i)_{i \in I'}$ of $\ker I_\pi$ such that the union of the family forms a basis of $\mathcal{H}_X$. We get a basis of $S_2(\mathcal{H}_X, \mathcal{H}_Y)$ through

$(d_j \otimes f_i)_{i \in I \cup I', j \in J}$ where $f_i = \mu_i^{1/2} e_i$ if $i \in I$ and $f_i = \bar{e}_i$ if $i \in I'$. By Equation (23) from [11], for $a > 0$ we then have

$$(C_{XX} + \lambda)^{-a} = \sum_{i \in I} (\mu_i + \lambda)^{-a} \left\langle \mu_i^{1/2} e_i, \cdot \right\rangle_{\mathcal{H}_X} \mu_i^{1/2} e_i + \lambda^{-a} \sum_{i \in I'} \langle \bar{e}_i, \cdot \rangle_{\mathcal{H}_X} \bar{e}_i.$$

Furthermore,

$$\begin{aligned}
C_{YX} &= \mathbb{E}_{YX} \left[ \phi_Y(Y) \otimes \phi_X(X) \right] \\
&= \mathbb{E}_X \left[ \mathbb{E}_{Y|X} \left[ \phi_Y(Y) \right] \otimes \phi_X(X) \right] \\
&= \mathbb{E}_X \left[ F_*(X) \otimes \phi_X(X) \right] \\
&= \mathbb{E}_X \left[ \Psi \left( C_{Y|X} \right) (X) \otimes \phi_X(X) \right] \\
&= \sum_{i \in I} \sum_{j \in J} a_{ij} \mathbb{E}_X \left[ \Psi \left( d_j \otimes [e_i] \right) (X) \otimes \phi_X(X) \right] \\
&= \sum_{i \in I} \sum_{j \in J} a_{ij} \mathbb{E}_X \left[ [e_i](X) d_j \otimes \phi_X(X) \right],
\end{aligned}$$

In the last step we used the explicit form of the isomorphism between $L_2(\pi; \mathcal{H}_Y)$ and $S_2(L_2(\pi), \mathcal{H}_Y)$ mentioned in Remark 1: $\Psi$ is characterized by $\Psi(g \otimes f) = (x \mapsto g f(x))$, for all $g \in \mathcal{H}_Y, f \in L_2(\pi)$. Then, using that $([e_i])_{i \in I}$ is an ONS in $L_2(\pi)$,

$$[C_{Y|X,\lambda}] = \sum_{i \in I} \sum_{j \in J} a_{ij} \frac{\mu_i}{\lambda + \mu_i} d_j \otimes [e_i],$$

and hence

$$[C_{Y|X,\lambda}] - C_{Y|X} = -\sum_{i \in I} \sum_{j \in J} a_{ij} \frac{\lambda}{\lambda + \mu_i} d_j \otimes [e_i]. \tag{15}$$

We are now ready to compute the upper bound. Parseval's identity w.r.t. the ONB $\left( d_j \otimes \mu_i^{\gamma/2} [e_i] \right)_{i \in I, j \in J}$ of $S_2 \left( [\mathcal{H}]_X^\gamma, \mathcal{H}_Y \right)$ yields

$$\begin{aligned}
\left\| [C_{Y|X,\lambda}] - C_{Y|X} \right\|_{S_2([\mathcal{H}]_X^\gamma, \mathcal{H}_Y)}^2 &= \left\| \sum_{i \in I} \sum_{j \in J} a_{ij} \frac{\lambda}{\lambda + \mu_i} d_j \otimes [e_i] \right\|_{S_2([\mathcal{H}]_X^\gamma, \mathcal{H}_Y)}^2 \\
&= \sum_{i \in I} \sum_{j \in J} a_{ij}^2 \left( \frac{\lambda}{\lambda + \mu_i} \right)^2 \mu_i^{-\gamma}.
\end{aligned}$$

Next we notice that,

$$\begin{aligned}
\left( \frac{\lambda}{\mu_i + \lambda} \right)^2 \mu_i^{-\gamma} &= \left( \frac{\lambda}{\mu_i + \lambda} \right)^2 \mu_i^{-\gamma} \left( \frac{\lambda}{\lambda} \frac{\mu_i + \lambda}{\mu_i + \lambda} \right)^{\beta - \gamma} \\
&= \lambda^{\beta - \gamma} \mu_i^{-\beta} \left( \frac{\lambda}{\mu_i + \lambda} \right)^2 \left( \frac{\mu_i}{\mu_i + \lambda} \right)^{\beta - \gamma} \left( \frac{\mu_i + \lambda}{\lambda} \right)^{\beta - \gamma} \\
&= \lambda^{\beta - \gamma} \mu_i^{-\beta} \left( \frac{\mu_i}{\mu_i + \lambda} \right)^{\beta - \gamma} \left( \frac{\lambda}{\lambda + \mu_i} \right)^{2 - \beta + \gamma} \\
&\leq \lambda^{\beta - \gamma} \mu_i^{-\beta},
\end{aligned}$$

where we used $\beta - \gamma \geq 0$ and $2 - \beta + \gamma \geq 0$. Hence,

$$\begin{aligned}
\left\| [C_{Y|X,\lambda}] - C_{Y|X} \right\|_{S_2([\mathcal{H}]_X^\gamma, \mathcal{H}_Y)}^2 &\leq \lambda^{\beta - \gamma} \sum_{i \in I} \sum_{j \in J} a_{ij}^2 \mu_i^{-\beta} \\
&= \lambda^{\beta - \gamma} \left\| C_{Y|X} \right\|_{S_2([\mathcal{H}]_X^\beta, \mathcal{H}_Y)}^2
\end{aligned}$$

$\square$

## A.2 Bounding the Variance

The proof will require several lemmas in its construction, which we now present. We start with a lemma that allows to go from the $\gamma$-norm of embedded vector-valued maps to their norm in the original Hilbert-Schmidt space.

**Lemma 2.** *For $0 \leq \gamma \leq 1$ and $F \in \mathcal{G}$ the inequality*

$$\|[F]\|_\gamma \leq \left\| CC_{XX}^{\frac{1-\gamma}{2}} \right\|_{S_2(\mathcal{H}_X, \mathcal{H}_Y)} \tag{16}$$

*holds, where $C = \bar{\Psi}^{-1}(F) \in S_2(\mathcal{H}_X, \mathcal{H}_Y)$. If, in addition, $\gamma < 1$ or $C \perp \mathcal{H}_Y \otimes \ker I_\pi$ is satisfied, then the result is an equality.*

*Proof.* Let us fix $F \in \mathcal{G}$, and define $C := \bar{\Psi}^{-1}(F) \in S_2(\mathcal{H}_X, \mathcal{H}_Y)$. Since $\left( \mu_i^{1/2} e_i \right)_{i \in I}$ is an ONB of $(\ker I_\pi)^\perp$, we can complete it with a basis $(\bar{e}_i)_{i \in I'}$ of $\ker I_\pi$ such that the union of the family forms a basis of $\mathcal{H}_X$. Let $(d_j)_{j \in J}$ be a basis of $\mathcal{H}_Y$, we get a basis of $S_2(\mathcal{H}_X, \mathcal{H}_Y)$ through $(d_j \otimes f_i)_{i \in I \cup I', j \in J}$ where $f_i = \mu_i^{1/2} e_i$ if $i \in I$ and $f_i = \bar{e}_i$ if $i \in I'$. Then $C$ admits the decomposition

$$C = \sum_{i \in I} \sum_{j \in J} a_{ij} d_j \otimes \mu_i^{1/2} e_i + \sum_{i \in I'} \sum_{j \in J} a_{ij} d_j \otimes \bar{e}_i,$$

where $a_{ij} = \langle C, d_j \otimes f_i \rangle_{S_2(\mathcal{H}_X, \mathcal{H}_Y)} = \langle C f_i, d_j \rangle_{\mathcal{H}_Y}$ for all $i \in I \cup I', j \in J$ (see [14]). Since

$$[C] = \sum_{i \in I} \sum_{j \in J} a_{ij} d_j \otimes \mu_i^{1/2} [e_i],$$

with Parseval's identity w.r.t. the ONB $\left( d_j \otimes \mu_i^{\gamma/2} [e_i] \right)_{i \in I \cup I', j \in J}$ of $S_2([\mathcal{H}]_X^\gamma, \mathcal{H}_Y)$ this yields

$$\|[C]\|_{S_2([\mathcal{H}]_X^\gamma, \mathcal{H}_Y)}^2 = \left\| \sum_{i \in I} \sum_{j \in J} a_{ij} \mu_i^{\frac{1-\gamma}{2}} d_j \otimes \mu_i^{\gamma/2} [e_i] \right\|_{S_2([\mathcal{H}]_X^\gamma, \mathcal{H}_Y)}^2 = \sum_{i \in I} \sum_{j \in J} a_{ij}^2 \mu_i^{1-\gamma}.$$

For $\gamma < 1$, the spectral decomposition of $C_{XX}$ together with the fact that $\left( d_j \otimes \mu_i^{1/2} e_i \right)_{i \in I, j \in J}$ is an ONS in $S_2(\mathcal{H}_X, \mathcal{H}_Y)$ yields

$$\left\| CC_{XX}^{\frac{1-\gamma}{2}} \right\|_{S_2(\mathcal{H}_X, \mathcal{H}_Y)}^2 = \left\| C \sum_{i \in I} \mu_i^{\frac{1-\gamma}{2}} \langle \cdot, \mu_i^{\frac{1}{2}} e_i \rangle_{\mathcal{H}_X} \mu_i^{\frac{1}{2}} e_i \right\|_{S_2(\mathcal{H}_X, \mathcal{H}_Y)}^2$$

$$= \sum_{i \in I} \left\| \sum_{l \in I} \mu_l^{\frac{1-\gamma}{2}} \langle \mu_i^{\frac{1}{2}} e_i, \mu_l^{\frac{1}{2}} e_l \rangle_{\mathcal{H}_X} \mu_l^{\frac{1}{2}} C e_l \right\|_{\mathcal{H}_Y}^2 + \sum_{i \in I'} \left\| \sum_{l \in I} \mu_l^{\frac{1-\gamma}{2}} \langle \bar{e}_i, \mu_l^{\frac{1}{2}} e_l \rangle_{\mathcal{H}_X} \mu_l^{\frac{1}{2}} C e_l \right\|_{\mathcal{H}_Y}^2$$

$$= \sum_{i \in I} \left\| \mu_i^{\frac{1-\gamma}{2}} \mu_i^{\frac{1}{2}} C e_i \right\|_{\mathcal{H}_Y}^2$$

$$= \sum_{i \in I} \sum_{j \in J} \mu_i^{1-\gamma} \left\langle C \left( \mu_i^{\frac{1}{2}} e_i \right), d_j \right\rangle_{\mathcal{H}_Y}^2$$

$$= \sum_{i \in I} \sum_{j \in J} a_{ij}^2 \mu_i^{1-\gamma}. \tag{17}$$

This proves the claimed equality in the case of $\gamma < 1$. For $\gamma = 1$, we have $C_{XX}^{\frac{1-\gamma}{2}} = \mathrm{Id}_{\mathcal{H}_X}$ and the Pythagorean theorem together with Parseval's identity yields

$$
\begin{aligned}
\left\| CC_{XX}^{\frac{1-\gamma}{2}} \right\|_{S_2(\mathcal{H}_X,\mathcal{H}_Y)}^2 &= \left\| \sum_{i\in I}\sum_{j\in J} a_{ij}d_j \otimes \mu_i^{1/2}e_i + \sum_{i\in I'}\sum_{j\in J} a_{ij}d_j \otimes \bar{e}_i \right\|_{S_2(\mathcal{H}_X,\mathcal{H}_Y)}^2 \\
&= \left\| \sum_{i\in I}\sum_{j\in J} a_{ij}d_j \otimes \mu_i^{1/2}e_i \right\|_{S_2(\mathcal{H}_X,\mathcal{H}_Y)}^2 + \left\| \sum_{i\in I'}\sum_{j\in J} a_{ij}d_j \otimes \bar{e}_i \right\|_{S_2(\mathcal{H}_X,\mathcal{H}_Y)}^2 \\
&= \sum_{i\in I}\sum_{j\in J} a_{ij}^2 + \left\| \sum_{i\in I'}\sum_{j\in J} a_{ij}d_j \otimes \bar{e}_i \right\|_{S_2(\mathcal{H}_X,\mathcal{H}_Y)}^2
\end{aligned}
\tag{18}
$$

This gives the claimed equality if $C \perp \mathcal{H}_Y \otimes \ker I_\pi$, as well as the claimed inequality for general $C \in S_2(\mathcal{H}_X,\mathcal{H}_Y)$. We conclude with $\|[F]\|_\gamma = \|[C]\|_{S_2([\mathcal{H}]_X^\gamma,\mathcal{H}_Y)}$ by definition. $\qquad\square$

**Lemma 3.** *If $F_* \in [\mathcal{G}]^\beta$ is satisfied for some $0 \le \beta \le 2$, then the following bounds are satisfied, for all $\lambda > 0$:*

$$
\|[F_\lambda] - F_*\|_\gamma^2 \le \|F_*\|_\beta^2 \lambda^{\beta-\gamma} \qquad (0 \le \gamma \le \beta),
\tag{19}
$$

$$
\|[F_\lambda]\|_\gamma^2 \le \|F_*\|_{\min\{\gamma,\beta\}}^2 \lambda^{-(\gamma-\beta)_+} \qquad (\gamma \ge 0).
\tag{20}
$$

*Proof.* The first term corresponds to the bias and has already been covered in Lemma 1. To show the second term, we get from Parseval's identity

$$
\|[F_\lambda]\|_\gamma^2 = \sum_{i\in I}\sum_{j\in J} \left( \frac{\mu_i}{\mu_i + \lambda} \right)^2 \mu_i^{-\gamma} a_{ij}^2.
$$

where $a_{ij} = \left\langle C_{Y|X}[e_i], d_j \right\rangle_{\mathcal{H}_Y}$ for all $i \in I, j \in J$ as in the proof of Lemma 1. In the case of $\gamma \le \beta$ we estimate the fraction by 1 and then Parseval's identity gives us

$$
\|[F_\lambda]\|_\gamma^2 \le \sum_{i\in I}\sum_{j\in J} \mu_i^{-\gamma} a_{ij}^2 = \|F_*\|_\gamma^2.
$$

In the case of $\gamma > \beta$,

$$
\|[F_\lambda]\|_\gamma^2 = \sum_{i\in I}\sum_{j\in J} \left( \frac{\mu_i^{1-\frac{\gamma-\beta}{2}}}{\mu_i + \lambda} \right)^2 \mu_i^{-\beta} a_{ij}^2 \le \lambda^{-(\gamma-\beta)} \sum_{i\in I}\sum_{j\in J} \mu_i^{-\beta} a_{ij}^2 = \lambda^{-(\gamma-\beta)} \|F_*\|_\beta^2,
$$

where we used Parseval's identity in the equality and Lemma 25 from [11].

$\qquad\square$

By (EMB), the inclusion map $I_\pi^{\alpha,\infty} : [\mathcal{H}]_X^\alpha \hookrightarrow L_\infty(\pi)$ has bounded norm $A > 0$ i.e. for $f \in [\mathcal{H}]_X^\alpha$, $f$ is $\pi-$a.e. bounded and $\|f\|_\infty \le A\|f\|_\alpha$. We know show that (EMB) automatically implies that the inclusion operator for $[\mathcal{G}]^\alpha$ is bounded.

**Lemma 4.** *Under (EMB) the inclusion operator $\mathcal{I}_\pi^{\alpha,\infty} : [\mathcal{G}]^\alpha \hookrightarrow L_\infty(\pi;\mathcal{H}_Y)$ is bounded with operator norm less than or equal to $A$.*

$L_\infty(\pi;\mathcal{H}_Y)$ denotes the space of $\mathcal{F}_{E_X} - \mathcal{F}_{\mathcal{H}_Y}$ measurable $\mathcal{H}_Y$-valued functions (gathered by $\pi$-equivalent classes) that are essentially bounded with respect to $\pi$. $L_\infty(\pi;\mathcal{H}_Y)$ is endowed with the norm $\|f\|_\infty := \inf\{c \ge 0 : \|f(x)\|_{\mathcal{H}_Y} \le c$ for $\pi$-almost every $x \in E_X\}$.

*Proof.* For every $F \in [\mathcal{G}]^\alpha$, there is a sequence $a_{ij} \in \ell_2(I \times J)$ such that for $\pi-$almost all $x \in E_X$,

$$
F(x) = \sum_{i\in I, j\in J} a_{ij}d_j \mu_i^{\alpha/2}[e_i](x)
$$

where $(d_j)_{j \in J}$ is any orthonormal basis of $\mathcal{H}_Y$ and $\|F\|_\alpha^2 = \sum_{i \in I, j \in J} a_{ij}^2$. We consider $F \in [\mathcal{G}]^\alpha$ such that $\sum_{i \in I, j \in J} a_{ij}^2 \leq 1$. For $\pi-$almost all $x \in E_X$,

$$\|F(x)\|_{\mathcal{H}_Y}^2 = \left\| \sum_{j \in J} \left( \sum_{i \in I} a_{ij} \mu_i^{\alpha/2}[e_i](x) \right) d_j \right\|_{\mathcal{H}_Y}^2$$

$$= \sum_{j \in J} \left( \sum_{i \in I} a_{ij} \mu_i^{\alpha/2}[e_i](x) \right)^2$$

$$\leq \sum_{j \in J} \left( \sum_{i \in I} a_{ij}^2 \sum_{i \in I} \mu_i^\alpha [e_i](x)^2 \right)$$

$$\leq A^2 \sum_{j \in J} \sum_{i \in I} a_{ij}^2$$

$$\leq A^2$$

where we used the Cauchy-Schwarz inequality for each $j \in J$ for the first inequality and a consequence of (EMB) in the second inequality (see Theorem 9 in [11]). We therefore conclude $\|\mathcal{I}_\pi^{\alpha,\infty}\| \leq A$. □

Combining Lemmas 3 and 4 we have the following corollary.

**Lemma 5.** *If $F_* \in [\mathcal{G}]^\beta$ and (EMB) are satisfied for some $0 \leq \beta \leq 2$ and $0 < \alpha \leq 1$, then the following bounds are satisfied, for all $0 < \lambda \leq 1$:*

$$\|[F_\lambda] - F_*\|_\infty^2 \leq (\|F_*\|_\infty + A\|F_*\|_\beta)^2 \lambda^{\beta - \alpha}, \tag{21}$$

$$\|[F_\lambda]\|_\infty^2 \leq A^2 \|F_*\|_{\min\{\alpha,\beta\}}^2 \lambda^{-(\alpha-\beta)_+}. \tag{22}$$

*In addition, we have $\|F_*\|_\infty \leq \kappa_Y$.*

*Proof.* For Eq. 22, we use Lemma 4 and Eq. 20 in Lemma 3,

$$\|[F_\lambda]\|_\infty^2 \leq A^2 \|[F_\lambda]\|_\alpha^2 \leq A^2 \|F_*\|_{\min\{\alpha,\beta\}}^2 \lambda^{-(\alpha-\beta)_+}$$

To show Eq. 21, in the case $\beta \leq \alpha$ we use the triangle inequality, Eq. 22 and $\lambda \leq 1$ to obtain

$$\|[F_\lambda] - F_*\|_\infty \leq \|F_*\|_\infty + \|[F_\lambda]\|_\infty$$
$$\leq \left( \|F_*\|_\infty + A\|F_*\|_\beta \right) \lambda^{-\frac{\alpha-\beta}{2}}$$

In the case $\beta > \alpha$, Eq. 21 is a consequence of Lemma 4 and Eq. 19 in Lemma 3 with $\gamma = \alpha$,

$$\|[F_\lambda] - F_*\|_\infty^2 \leq A^2 \|[F_\lambda] - F_*\|_\alpha^2 \leq A^2 \|F_*\|_\beta^2 \lambda^{\beta-\alpha} \leq (\|F_*\|_\infty + A\|F_*\|_\beta)^2 \lambda^{\beta-\alpha}.$$

We emphasize that $F_*$ always belongs to $L_\infty(\pi; \mathcal{H}_Y)$. Indeed, for $\pi$-almost all $x \in E_X$ we have

$$\|F_*(x)\|_{\mathcal{H}_Y} = \left\| \int_{E_X} \phi_Y(y) p(x, dy) \right\|_{\mathcal{H}_Y}$$

$$\leq \int_{E_X} \|\phi_Y(y)\|_{\mathcal{H}_Y} p(x, dy)$$

$$\leq \int_{E_X} \kappa_Y p(x, dy) = \kappa_Y.$$

□

**Theorem 4.** *Suppose Assumptions 1 to 3 and* ([EMB](#)) *with $A > 0$ hold. We define*

$$
\begin{aligned}
M(\lambda) &= \left\| [F_\lambda] - F_* \right\|_\infty, \\
\mathcal{N}(\lambda) &= \operatorname{tr}\left( C_{XX} \left( C_{XX} + \lambda \right)^{-1} \right), \\
Q_\lambda &= \max\{ M(\lambda), 2\kappa_Y \}, \\
g_\lambda &= \log\left( 2e\mathcal{N}(\lambda) \frac{\|C_{XX}\| + \lambda}{\|C_{XX}\|} \right).
\end{aligned}
$$

*Then, for $\tau \geq 1$, $\lambda > 0$, $n \geq 8A^2 \tau g_\lambda \lambda^{-\alpha}$ and $\lambda > 0$, with probability $1 - 4e^{-\tau}$ :*

$$
\left\| \left[ \hat{C}_{Y|X,\lambda} - C_{Y|X,\lambda} \right] \right\|^2_{S_2\left( [\mathcal{H}]_X^\gamma, \mathcal{H}_Y \right)}
$$
$$
\leq \frac{576\tau^2}{n\lambda^\gamma} \left( 4\kappa_Y^2 \mathcal{N}(\lambda) + \frac{\|F_* - [F_\lambda]\|^2_{L_2(\pi; \mathcal{H}_Y)} A^2}{\lambda^\alpha} + \frac{2Q_\lambda^2 A^2}{n\lambda^\alpha} \right) \tag{23}
$$

*Proof.* We first decompose the variance term as

$$
\left\| \left[ \hat{C}_{Y|X,\lambda} - C_{Y|X,\lambda} \right] \right\|_{S_2\left( [\mathcal{H}]_X^\gamma, \mathcal{H}_Y \right)} \tag{24}
$$
$$
= \left\| \left[ \hat{C}_{YX} \left( \hat{C}_{XX} + \lambda Id \right)^{-1} - C_{YX} \left( C_{XX} + \lambda Id \right)^{-1} \right] \right\|_{S_2\left( [\mathcal{H}]_X^\gamma, \mathcal{H}_Y \right)}
$$
$$
\leq \left\| \left( \hat{C}_{YX} \left( \hat{C}_{XX} + \lambda Id \right)^{-1} - C_{YX} \left( C_{XX} + \lambda Id \right)^{-1} \right) C_{XX}^{\frac{1-\gamma}{2}} \right\|_{S_2(\mathcal{H}_X, \mathcal{H}_Y)}
$$
$$
\leq \left\| \left( \hat{C}_{YX} - C_{YX} \left( C_{XX} + \lambda Id \right)^{-1} \left( \hat{C}_{XX} + \lambda Id \right) \right) \left( C_{XX} + \lambda Id \right)^{-\frac{1}{2}} \right\|_{S_2(\mathcal{H}_X, \mathcal{H}_Y)} \tag{25}
$$
$$
\cdot \left\| \left( C_{XX} + \lambda Id \right)^{\frac{1}{2}} \left( \hat{C}_{XX} + \lambda Id \right)^{-1} \left( C_{XX} + \lambda Id \right)^{\frac{1}{2}} \right\|_{\mathcal{H}_X \to \mathcal{H}_X} \tag{26}
$$
$$
\cdot \left\| \left( C_{XX} + \lambda Id \right)^{-\frac{1}{2}} C_{XX}^{\frac{1-\gamma}{2}} \right\|_{\mathcal{H}_X \to \mathcal{H}_X} \tag{27}
$$

where we used Lemma [2](#). Eq. ([26](#)) is bounded as in Theorem 16 in [[11]](#),

$$
\left\| \left( C_{XX} + \lambda Id \right)^{\frac{1}{2}} \left( \hat{C}_{XX} + \lambda Id \right)^{-1} \left( C_{XX} + \lambda Id \right)^{\frac{1}{2}} \right\| \leq 3
$$

for $n \geq 8A^2 \tau g_\lambda \lambda^{-\alpha}$ with probability $1 - 2e^{-\tau}$. For Eq. ([27](#)) we have, using Lemma 25 from [[11]](#)

$$
\left\| \left( C_{XX} + \lambda Id \right)^{-\frac{1}{2}} C_{XX}^{\frac{1-\gamma}{2}} \right\| \leq \sqrt{\sup_i \frac{\mu_i^{1-\gamma}}{\mu_i + \lambda}} \leq \lambda^{-\frac{\gamma}{2}}.
$$

Finally for the bound of Eq. ([25](#)) we show that for $\tau \geq 1$, $\lambda > 0$ and $n \geq 1$ with probability $1 - 2e^{-\tau}$:

$$
\left\| \left( \hat{C}_{YX} - C_{YX} \left( C_{XX} + \lambda Id \right)^{-1} \left( \hat{C}_{XX} + \lambda Id \right) \right) \left( C_{XX} + \lambda Id \right)^{-\frac{1}{2}} \right\|^2_{S_2(\mathcal{H}_X, \mathcal{H}_Y)}
$$
$$
\leq \frac{64\tau^2}{n} \left( 4\kappa_Y^2 \mathcal{N}(\lambda) + \frac{\|F_* - [F_\lambda]\|^2_{L_2(\pi; \mathcal{H}_Y)} A^2}{\lambda^\alpha} + \frac{2Q_\lambda^2 A^2}{n\lambda^\alpha} \right). \tag{28}
$$

We begin with the decomposition

$$\hat{C}_{YX} - C_{YX}(C_{XX} + \lambda Id)^{-1}\left(\hat{C}_{XX} + \lambda Id\right)$$

$$= \hat{C}_{YX} - C_{YX}(C_{XX} + \lambda Id)^{-1}\left(C_{XX} + \lambda Id + \hat{C}_{XX} - C_{XX}\right)$$

$$= \hat{C}_{YX} - C_{YX} + C_{YX}(C_{XX} + \lambda Id)^{-1}\left(C_{XX} - \hat{C}_{XX}\right)$$

$$= \hat{C}_{YX} - C_{YX}(C_{XX} + \lambda Id)^{-1}\hat{C}_{XX} - \left(C_{YX} - C_{YX}(C_{XX} + \lambda Id)^{-1}C_{XX}\right)$$

$$= \hat{C}_{YX} - C_{YX}(C_{XX} + \lambda Id)^{-1}\hat{\mathbb{E}}[\phi_X(X) \otimes \phi_X(X)] - \left(C_{YX} - C_{YX}(C_{XX} + \lambda Id)^{-1}\mathbb{E}[\phi_X(X) \otimes \phi_X(X)]\right)$$

$$= \hat{\mathbb{E}}\left[(\phi_Y(Y) - F_\lambda(X)) \otimes \phi_X(X)\right] - \mathbb{E}\left[(\phi_Y(Y) - F_\lambda(X)) \otimes \phi_X(X)\right]$$

where we denote $\hat{\mathbb{E}}[\phi_X(X) \otimes \phi_X(X)] = \frac{1}{n}\sum_i^n \phi_X(x_i) \otimes \phi_X(x_i)$. We wish to apply Theorem 6 with $H = S_2(\mathcal{H}_X, \mathcal{H}_Y)$. We emphasise the difference from [41], where the proof is formulated for bounded linear operators. Consider the random variables $\xi_0, \xi_2 : E_X \times E_Y \to \mathcal{H}_Y \otimes \mathcal{H}_X$ defined by

$$\xi_0(x, y) := (\phi_Y(y) - F_\lambda(x)) \otimes \phi_X(x),$$

$$\xi_2(x, y) := \xi_0(x, y)(C_{XX} + \lambda Id)^{-1/2}.$$

Moreover, since our kernels $k_X$ and $k_Y$ are bounded,

$$\|\xi_0(x, y)\|_{S_2(\mathcal{H}_X, \mathcal{H}_Y)} = \|\phi_Y(y) - F_\lambda(x)\|_{\mathcal{H}_Y}\|\phi_X(x)\|_{\mathcal{H}_X}$$

$$\leq \|\phi_Y(y) - F_\lambda(x)\|_{\mathcal{H}_Y}\kappa_X$$

$$\leq \left(\kappa_Y + \|F_\lambda(x)\|_{\mathcal{H}_Y}\right)\kappa_X,$$

and $F_\lambda$ is $\pi$-almost surely bounded by Lemma 5. As a result $\xi_0$ is Bochner-integrable. This yields

$$\frac{1}{n}\sum_{i=1}^n (\xi_2(x_i, y_i) - \mathbb{E}\xi_2) = \hat{\mathbb{E}}\xi_2 - \mathbb{E}\xi_2 = \left(\hat{C}_{YX} - C_{YX}(C_{XX} + \lambda Id)^{-1}\left(\hat{C}_{XX} + \lambda Id\right)\right)(C_{XX} + \lambda Id)^{-\frac{1}{2}},$$

and therefore Eq. (25) coincides with the left hand side of Bernstein's inequality for $H$-valued random variables (Theorem 6). Consequently, it remains to bound the $m$-th moment of $\xi_2$, for $m \geq 2$,

$$\mathbb{E}\|\xi_2\|_{S_2(\mathcal{H}_X, \mathcal{H}_Y)}^m = \int_{E_X}\left\|(C_{XX} + \lambda Id)^{-1/2}\phi(x)\right\|_{\mathcal{H}_X}^m \int_{E_Y}\|\phi_Y(y) - F_\lambda(x)\|_{\mathcal{H}_Y}^m\, p(x,\, \mathrm{d}y)\mathrm{d}\pi(x).$$

First, we consider the inner integral. Using the triangle inequality and the fact that $\|\phi_Y(y) - F_\lambda(x)\|_{\mathcal{H}_Y} \leq 2\kappa_Y$ almost surely,

$$\int_{E_Y}\|\phi_Y(y) - F_\lambda(x)\|_{\mathcal{H}_Y}^m\, p(x,\, \mathrm{d}y) \leq 2^{m-1}\left(\|\phi_Y(\cdot) - F_*(x)\|_{L_m(p(x,\cdot))}^m + \|F_*(x) - F_\lambda(x)\|_{\mathcal{H}_Y}^m\right)$$

$$\leq 2^{2m-1}\kappa_Y^m + 2^{m-1}\|F_*(x) - F_\lambda(x)\|_{\mathcal{H}_Y}^m$$

for $\pi$-almost all $x \in E_X$. If we plug this bound into the outer integral and use the abbreviation $h_x := (C_{XX} + \lambda)^{-1/2}\phi_X(\cdot)$ we get

$$\mathbb{E}\|\xi_2\|_{S_2(\mathcal{H}_X, \mathcal{H}_Y)}^m \leq 2^{2m-1}\kappa_Y^m\int_{E_X}\|h_x\|_{\mathcal{H}_X}^m\, \mathrm{d}\pi(x) + 2^{m-1}\int_{E_X}\|h_x\|_{\mathcal{H}_X}^m\|F_*(x) - F_\lambda(x)\|_{\mathcal{H}_Y}^m\, \mathrm{d}\pi(x). \tag{29}$$

Using Lemma 13 [11], we can bound the first term in Equation 29 above by

$$2^{2m-1}\kappa_Y^m\int_{E_X}\|h_x\|_{\mathcal{H}_X}^m\, \mathrm{d}\pi(x) \leq 2^{2m-1}\kappa_Y^m\left(\frac{A}{\lambda^{\alpha/2}}\right)^{m-2}\int_{E_X}\|h_x\|_{\mathcal{H}_X}^2\, \mathrm{d}\pi(x)$$

$$= \left(\frac{4\kappa_Y A}{\lambda^{\alpha/2}}\right)^{m-2}8\kappa_Y^2\mathcal{N}(\lambda)$$

$$\leq \frac{1}{2}m!\left(\frac{2Q_\lambda A}{\lambda^{\alpha/2}}\right)^{m-2}8\kappa_Y^2\mathcal{N}(\lambda)$$

where we only used $2\kappa_Y \leq Q_\lambda$ and $\frac{1}{2}m! \geq 1$ in the last step. Again, using Lemma 13 from [11], the second term in Equation (29) can be bounded by

$$2^{m-1} \int_{E_X} \|h_x\|_{\mathcal{H}_X}^m \|F_*(x) - F_\lambda(x)\|_{\mathcal{H}_Y}^m \ \mathrm{d}\pi(x)$$

$$\leq \frac{1}{2} \left( \frac{2A}{\lambda^{\alpha/2}} \right)^m M(\lambda)^{m-2} \int_{E_X} \|F_*(x) - F_\lambda(x)\|_{\mathcal{H}_Y}^2 \ \mathrm{d}\pi(x)$$

$$= \frac{1}{2} \left( \frac{2AM(\lambda)}{\lambda^{\alpha/2}} \right)^{m-2} \|F_* - [F_\lambda]\|_{L_2(\pi;\mathcal{H}_Y)}^2 \frac{4A^2}{\lambda^\alpha}$$

$$\leq \frac{1}{2} m! \left( \frac{2Q_\lambda A}{\lambda^{\alpha/2}} \right)^{m-2} \|F_* - [F_\lambda]\|_{L_2(\pi;\mathcal{H}_Y)}^2 \frac{2A^2}{\lambda^\alpha},$$

where we only used $M(\lambda) \leq Q_\lambda$ and $2 \leq m$ ! in the last step. Finally, we get

$$\mathbb{E} \|\xi_2\|_{S_2(\mathcal{H}_X,\mathcal{H}_Y)}^m \leq \frac{1}{2} m! \left( \frac{2Q_\lambda A}{\lambda^{\alpha/2}} \right)^{m-2} 2 \left( 4\kappa_Y^2 \mathcal{N}(\lambda) + \|F_* - [F_\lambda]\|_{L_2(\pi;\mathcal{H}_Y)}^2 \frac{A^2}{\lambda^\alpha} \right)$$

and an application of Bernstein's inequality from Theorem 6 with $L = 2Q_\lambda A\lambda^{-\alpha/2}$ and $\sigma^2 = 2 \left( 4\kappa_Y^2 \mathcal{N}(\lambda) + \|F_* - [F_\lambda]\|_{L_2(\pi;\mathcal{H}_Y)}^2 A^2\lambda^{-\alpha} \right)$ yield the bound in Eq. 28. Putting all the terms together, we obtain our result. $\qquad \square$

### A.3 The CME Learning Rate

In this section, we aim to establish our upper bound on the learning rate of the conditional mean embedding by combining the learning rates obtained for the bias and variance.

Let us fix some $\tau \geq 1$ and a lower bound $0 < c_1 \leq 1$ with $c_1 \leq \|C_{XX}\|$. We first show that Theorem 4 is applicable. To this end, we prove that there is an index bound $n_0 \geq 1$ such that $n \geq 8A^2 \log \tau g_{\lambda_n} \lambda_n^{-\alpha}$ is satisfied for all $n \geq n_0$. Since $\lambda_n \to 0$ we choose $n_0' \geq 1$ such that $\lambda_n \leq c_1 \leq \min\{1, \|C_{XX}\|\}$ for all $n \geq n_0'$. We get for $n \geq n_0'$,

$$\frac{8A^2 \tau g_{\lambda_n} \lambda_n^{-\alpha}}{n} = \frac{8A^2 \tau \lambda_n^{-\alpha}}{n} \cdot \log \left( 2e\mathcal{N}(\lambda_n) \frac{\|C_{XX}\| + \lambda_n}{\|C_{XX}\|} \right)$$

$$\leq \frac{8A^2 \tau \lambda_n^{-\alpha}}{n} \cdot \log \left( 4ce\lambda_n^{-p} \right)$$

$$= 8A^2 \tau \left( \frac{\log (4ce) \lambda_n^{-\alpha}}{n} + \frac{p\lambda_n^{-\alpha} \log \lambda_n^{-1}}{n} \right)$$

where the second step uses Lemma 10. Hence, it is enough to show $\frac{\log(\lambda_n^{-1})}{n\lambda_n^\alpha} \to 0$. We consider the cases $\beta + p \leq \alpha$ and $\beta + p > \alpha$.

- $\beta + p \leq \alpha$. By substituting that $\lambda_n = \Theta \left( \left( \frac{n}{\log^r n} \right)^{-\frac{1}{\alpha}} \right)$ for some $r > 1$ we have

$$\frac{\lambda_n^{-\alpha} \log \lambda_n^{-1}}{n} = \Theta \left( \frac{\log(n)}{n} \frac{n}{\log^r(n)} \right) = \Theta \left( \frac{1}{\log^{r-1}(n)} \right) \to 0, \text{ as } n \to \infty.$$

- $\beta + p > \alpha$. By substituting that $\lambda_n = \Theta \left( n^{-\frac{1}{\beta+p}} \right)$ and using $1 - \frac{\alpha}{\beta+p} > 0$ we have

$$\frac{\lambda_n^{-\alpha} \log \lambda_n^{-1}}{n} = \Theta \left( \frac{\log(n)}{n} n^{\frac{\alpha}{\beta+p}} \right) = \Theta \left( \frac{\log(n)}{n^{1-\frac{\alpha}{\beta+p}}} \right) \to 0, \text{ as } n \to \infty.$$

Consequently, there is a $n_0 \geq n_0'$ with $n \geq 8A^2 \log \tau g_{\lambda_n} \lambda_n^{-\alpha}$ for all $n \geq n_0$. Moreover, $n_0$ just depends on $\lambda_n, c, c_1, \tau, A$, and on the parameters $\alpha, p$.

Let $n \geq n_0$ be fixed. By Theorem 4, we have

$$\left\| \left[ \hat{C}_{Y|X,\lambda} - C_{Y|X,\lambda} \right] \right\|_{S_2([\mathcal{H}]_X^\gamma, \mathcal{H}_Y)}^2 \leq \frac{576\tau^2}{n\lambda_n^\gamma} \left( 4\kappa_Y^2 \mathcal{N}(\lambda_n) + \frac{\|F_* - [F_\lambda]\|_{L_2(\pi;\mathcal{H}_Y)}^2 A^2}{\lambda_n^\alpha} + \frac{2Q_{\lambda_n}^2 A^2}{n\lambda_n^\alpha} \right).$$

Using Lemma 10, Lemma 3 with $\gamma = 0$, we have

$$\left\| [\hat{C}_{Y|X} - C_{Y|X}^\lambda] \right\|_{S_2([\mathcal{H}]_X^\gamma, \mathcal{H}_Y)}^2 \leq \frac{576\tau^2}{n\lambda_n^\gamma} \left( 4\kappa_Y^2 c\lambda_n^{-p} + A^2 \|F_*\|_\beta^2 \lambda_n^{\beta - \alpha} + \frac{2Q_{\lambda_n}^2 A^2}{n\lambda_n^\alpha} \right)$$

For the last term, using the definition of $Q_\lambda$ in Theorem 4 with Lemma 5 and $\lambda_n \leq 1$, we get

$$\begin{aligned}
Q_{\lambda_n}^2 &= \max\{(2\kappa_Y)^2, \|[F_\lambda] - F_*\|_\infty^2\} \\
&\leq \max\left\{ (2\kappa_Y)^2, (\|F_*\|_\infty + A\|F_*\|_\beta)^2 \lambda_n^{-(\alpha - \beta)} \right\} \\
&\leq K_0 \lambda_n^{-(\alpha - \beta)_+},
\end{aligned} \tag{30}$$

where $K_0 := \max\left\{ (2\kappa_Y)^2, (B_\infty + A\|F_*\|_\beta)^2 \right\}$. Thus,

$$\left\| [\hat{C}_{Y|X} - C_{Y|X}^\lambda] \right\|_{S_2([\mathcal{H}]_X^\gamma, \mathcal{H}_Y)}^2 \leq \frac{576\tau^2}{n\lambda_n^\gamma} \left( 4\kappa_Y^2 c\lambda_n^{-p} + A^2 \|F_*\|_\beta^2 \lambda_n^{\beta - \alpha} + 2A^2 K_0 \frac{1}{n\lambda_n^{\alpha + (\alpha - \beta)_+}} \right).$$

For the first and second terms in the bracket, we use again the fact that $\lambda_n \leq 1$, and get

$$4c\kappa_Y^2 \lambda_n^{-p} + A^2 \|F_*\|_\beta^2 \lambda_n^{-(\alpha - \beta)} \leq \left( 4c\kappa_Y^2 + A^2 \|F_*\|_\beta^2 \right) \max\{\lambda_n^{-p}, \lambda_n^{-(\alpha - \beta)}\} \leq K_1 \lambda_n^{-\max\{p, \alpha - \beta\}}$$

with $K_1 := 4c\kappa_Y^2 + A^2 \|F_*\|_\beta^2$. We now have

$$\begin{aligned}
\left\| [\hat{C}_{Y|X} - C_{Y|X}^\lambda] \right\|_{S_2([\mathcal{H}]_X^\gamma, \mathcal{H}_Y)}^2 &\leq \frac{576\tau^2}{n\lambda_n^\gamma} \left( K_1 \lambda_n^{-\max\{p, \alpha - \beta\}} + 2A^2 K_0 \frac{1}{n\lambda_n^{\alpha + (\alpha - \beta)_+}} \right) \\
&= \frac{576\tau^2}{n\lambda_n^{\gamma + \max\{p, \alpha - \beta\}}} \left( K_1 + 2A^2 K_0 \frac{1}{n\lambda_n^{\alpha + (\alpha - \beta)_+ - \max\{p, \alpha - \beta\}}} \right).
\end{aligned}$$

Again, we treat the cases $\beta + p \leq \alpha$ and $\beta + p > \alpha$ separately.

- $\beta + p \leq \alpha$. In this case we have
$$\alpha + (\alpha - \beta)_+ - \max\{p, \alpha - \beta\} = \alpha.$$
Since $\lambda_n = \Theta\left( \left( \frac{n}{\log^r n} \right)^{-\frac{1}{\alpha}} \right)$, for some $r > 1$ we therefore have
$$\frac{1}{n\lambda_n^{\alpha + (\alpha - \beta)_+ - \max\{p, \alpha - \beta\}}} = \frac{1}{n\lambda^\alpha} = \Theta\left( \frac{1}{\log^r n} \right).$$

- $\beta + p > \alpha$. We have $p > \alpha - \beta$ and $\lambda_n = \Theta\left( n^{-\frac{1}{\beta + p}} \right)$, and hence
$$\frac{1}{n\lambda_n^{\alpha + (\alpha - \beta)_+ - \max\{p, \alpha - \beta\}}} = \frac{1}{n\lambda_n^{\alpha + (\alpha - \beta)_+ - p}} = \Theta\left( \left( \frac{1}{n} \right)^{1 - \frac{\alpha + (\alpha - \beta)_+ - p}{\beta + p}} \right).$$
Using $p > \alpha - \beta$ again gives us
$$1 - \frac{\alpha + (\alpha - \beta)_+ - p}{\beta + p} = \frac{2p - (\alpha - \beta)_+ - (\alpha - \beta)}{\beta + p} > 0.$$

As such, there is a constant $K_2 > 0$ with

$$\left\| [\hat{F}_\lambda - F_\lambda] \right\|_\gamma^2 = \left\| [\hat{C}_{Y|X} - C_{Y|X}^\lambda] \right\|_{S_2([\mathcal{H}]_X^\gamma, \mathcal{H}_Y)}^2 \leq 576 \frac{\tau^2}{n\lambda_n^{\gamma + \max\{p, \alpha - \beta\}}} \left( K_1 + 2A^2 K_0 K_2 \right)$$

for all $n \geq n_0$. Defining $K_3 := 576(K_1 + 2A^2 K_0 K_2)$, and using the bias-variance splitting from Eq. (13) and Lemma 1, we have

$$\begin{aligned}
\left\| [\hat{F}_\lambda] - F_* \right\|_\gamma^2 &\leq 2\|C_{Y|X}\|_{S_2([\mathcal{H}]_X^\beta, \mathcal{H}_Y)}^2 \lambda_n^{\beta - \gamma} + 2K_3 \frac{\tau^2}{n\lambda_n^{\gamma + \max\{p, \alpha - \beta\}}} \\
&\leq \tau^2 \lambda_n^{\beta - \gamma} \left( 2\|C_{Y|X}\|_{S_2([\mathcal{H}]_X^\beta, \mathcal{H}_Y)}^2 + 2K_3 \frac{1}{n\lambda_n^{\max\{\beta + p, \alpha\}}} \right),
\end{aligned}$$

where we used $\tau \geq 1$ and $\lambda_n \leq 1$. Since in both cases $\beta + p \leq \alpha$ and $\beta + p > \alpha$, $\lambda_n \succcurlyeq n^{-1/\max\{\alpha, \beta + p\}}$ there is some constant $K > 0$ such that

$$\left\| [\hat{F}_\lambda] - F_* \right\|_\gamma^2 \leq \tau^2 K \lambda_n^{\beta - \gamma}$$

for all $n \geq n_0$.

# B   Proof of Theorem 3

In this section, we establish a lower bound on the learning rate for the empirical conditional mean embedding. To this end, we build on the lower bound for kernel ridge regression for real-valued outputs in [11], and for finite dimensional vector-valued outputs in [4, 15, 10]. The usual approach to build such lower bounds is to construct a family of distributions on the data space and to control the Kullback-Leibler divergence between each pair of distributions. We cannot directly adapt the proofs of [4, 15, 10] and [11], however, since both [15] and [11] requires the output space to be finite dimensional, which is not the case in our setting. In addition, [11] builds a Gaussian distribution for $Y$ conditioned on $X$. It would be a challenge to build a distribution on $E_X \times E_Y$ so as to attain the required Gaussian conditional distribution in feature space $\mathcal{H}_Y$, however.

Our novelty in obtaining the lower bound is to reduce the infinite dimensional learning to a specially designed scalar regression problem. We show that the learning risk is lower bounded by the learning problem evaluated at a particular point (Eq. (35)), which can be seen as the risk of a scalar-valued regression problem. This effectively allows us to derive the lower bound exploiting proof techniques from [4, 11].

We start by noticing that for any $F \in L_2(\pi; \mathcal{H}_Y)$ and $a \in E_Y$,

$$\int_{E_X} \left( \langle F(x), \phi_Y(a) \rangle_{\mathcal{H}_Y} - \langle F_*(x), \phi_Y(a) \rangle_{\mathcal{H}_Y} \right)^2 d\pi(x) \leq \int_{E_X} \|F(x) - F_*(x)\|_{\mathcal{H}_Y}^2 \|\phi_Y(a)\|_{\mathcal{H}_Y}^2 d\pi(x)$$
$$\leq \kappa_Y^2 \|F - F_*\|_{L_2(\pi; \mathcal{H}_Y)}^2. \tag{31}$$

Moreover, by Lemma 6, the inequality holds for general $\gamma$-norm (which implies the previous equation, setting $\gamma = 0$),

$$\|\langle F, \phi_Y(a) \rangle_{\mathcal{H}_Y} - \langle F_*, \phi_Y(a) \rangle_{\mathcal{H}_Y}\|_\gamma \leq \kappa_Y \|F - F_*\|_\gamma. \tag{32}$$

**Lemma 6.** *Let $\gamma \geq 0$, for any $F \in [\mathcal{G}]^\gamma$ and $a \in E_Y$, we have*

$$\|\langle F, \phi_Y(a) \rangle_{\mathcal{H}_Y}\|_\gamma \leq \kappa_Y \|F\|_\gamma.$$

*Proof.* The case where $\gamma = 0$ is already proved in Eq. (31). We now let $\gamma > 0$. Recall $\{d_j\}_{j \in J}$ and $\{\mu_i^{\gamma/2}[e_i]\}_{i \in I}$ are the orthonormal basis of $\mathcal{H}_Y$ and $[\mathcal{H}]_X^\gamma$, since $F \in [\mathcal{G}]^\gamma$, we can write $F$ as

$$F = \sum_{i,j} a_{ij} d_j \mu_i^{\gamma/2}[e_i].$$

Therefore, we have

$$\langle F(.), \phi_Y(a) \rangle_{\mathcal{H}_Y} = \sum_{i,j} a_{ij} d_j(a) \mu_i^{\gamma/2}[e_i](.).$$

$\langle F(.), \phi_Y(a) \rangle_{\mathcal{H}_Y}$ is a function in $[\mathcal{H}]_X^\gamma$ as

$$\|\langle F(.), \phi_Y(a) \rangle_{\mathcal{H}_Y}\|_\gamma^2 = \sum_i \left( \sum_j a_{ij} d_j(a) \right)^2$$
$$\leq \sum_i \sum_j a_{ij}^2 \sum_j d_j^2(a)$$
$$= k_Y(a, a) \sum_{i,j} a_{ij}^2$$
$$\leq \kappa_Y^2 \|F\|_\gamma^2 < +\infty,$$

where for the second step, we used Cauchy-Schwartz inequality. The third step is due to Parseval's identity since $\{d_j\}_{j \in J}$ is an orthonormal basis of $\mathcal{H}_Y$. $\qquad\square$

We now express the l.h.s as the risk of a scalar-valued regression. Consider a distribution $P$ on $E_X \times E_Y$ that factorizes as $P(x, y) = p(y \mid x)\pi(x)$ for all $(x, y) \in E_X \times E_Y$. For all $x \in E_X$, $p(\cdot \mid x)$ defines a probability distribution on $E_Y$. We fix an element $a \in E_Y$ and define $E_Y^a :=$

$k_Y(E_Y, a) = \{y_a \in \mathbb{R} \mid y_a = k_Y(y, a), y \in E_Y\}$. Consider the joint distribution $P_a$ on $E_X \times E_Y^a$ such that

$$p_a(. \mid x) := (k_Y(\cdot, a))_\# \, p(\cdot \mid x)$$
$$P_a(x, y_a) := p_a(y_a \mid x)\pi(x), \quad (x, y_a) \in E_X \times E_Y^a \tag{33}$$

where $\#$ denotes the push-forward operation. For a dataset $D = \{(x_i, y_i)\}_{i=1}^n \in (E_X \times E_Y)^n$ where the data are i.i.d from $P$, the dataset $D_a = \{(x_i, y_{a.i})\}_{i=1}^n \in (E_X \times E_Y^a)^n \subseteq (E_X \times \mathbb{R})^n$ where $y_{a.i} := k_Y(y_i, a)$ for all $i = 1, \ldots, n$ is i.i.d from $P_a$. Note that $p_a(\cdot \mid x)$ is a probability distribution on $\mathbb{R}$ for all $x$ supported by $\pi$. By definition of the push-forward operator, the Bayes predictor associated to the joint distribution $P_a$ is

$$f_{a,*}(x) = \int_{\mathbb{R}} y_a dp_a(y_a \mid x) = \int_{E_Y} k_Y(y, a) dp(y \mid x)$$
$$= \left\langle \int_{E_Y} \phi_Y(y) dp(y \mid x), \phi_Y(a) \right\rangle_{\mathcal{H}_Y} \tag{34}$$
$$= \langle F_*(x), \phi_Y(a) \rangle_{\mathcal{H}_Y}$$

where $F_*$ is the $\mathcal{H}_Y$-valued conditional mean embedding associated to $P$. Therefore, plugging Eq. (34) in Eq. (31) we obtain that for any learning method $D \to \hat{F}_D \in (\mathcal{H}_Y)^{E_X}$

$$\|[\hat{F}_D] - F_*\|_{L_2(\pi;\mathcal{H}_Y)} \geq \kappa_Y^{-1} \|[\hat{f}_{D_a}] - f_{a.*}\|_{L_2(\pi)} \tag{35}$$

where $\hat{f}_{D_a}(.) := \langle \hat{F}_D(.), \phi_Y(a) \rangle_{\mathcal{H}_Y}$. The r.h.s is the error measured in $L_2$-norm of the learning method $D_a \to \hat{f}_{D_a} \in \mathbb{R}^{E_X}$ on the scalar-regression learning problem associated to $D_a$.

To derive a lower bound on the r.h.s in Eq. 35, the strategy is to define a conditional distribution $p_a(. \mid x)$ on $E_Y^a$, $x \in E_X$, that is difficult to learn. As $E_Y^a$ is a bounded subset of $\mathbb{R}$, we cannot directly exploit the Gaussian conditional distributions used in [11]. Indeed, for all $y \in E_Y$, $|k_Y(y, a)| \leq \kappa_Y^2$. Instead, we suggest to swap the Gaussian conditional distributions used in [11] with the discrete conditional distributions used in [4, 15, 10].

We will need the following Lemma that corresponds to Lemma 19, Lemma 23 and Equation (55) in [11].

**Lemma 7.** *Let $k_X$ be a kernel on $E_X$ such that Assumptions 1-3 hold and $\pi$ be a probability distribution on $E_X$ such that (EVD+) and (EMB) are satisfied for some $0 < p \leq \alpha \leq 1$. Then, for all parameters $0 < \beta \leq 2$, $0 \leq \gamma \leq 1$ with $\gamma < \beta$ and all constants $\bar{B}, B_\infty > 0$, there exist constants $0 < \epsilon_0 \leq 1$ and $C_0, C > 0$ such that the following statement is satisfied: for all $0 < \epsilon \leq \epsilon_0$ there is an $M_\epsilon \geq 1$ with*

$$2^{C_0 \epsilon^{-u}} \leq M_\epsilon \leq 2^{3C_0 \epsilon^{-u}} \tag{36}$$

*where $u := \frac{p}{\max\{\alpha, \beta\} - \gamma}$ and functions $f_1, \ldots, f_{M_\epsilon}$ such that $f_i \in [\mathcal{H}]_X^\beta$, $\|f_i\|_\beta \leq \bar{B}$, $\|f_i\|_{L_\infty(\pi)} \leq B_\infty$, and*

$$\|f_i - f_j\|_\gamma^2 \geq 4\epsilon \tag{37}$$
$$\|f_i - f_j\|_{L_2(\pi)}^2 \leq 32C^\gamma \epsilon m^{-\gamma/p}, \tag{38}$$

*for all $i, j \in \{0, \ldots, M_\varepsilon\}$ with $i \neq j$ where $m$ comes from Lemma 23 in [11].*

We now combine Lemma 7 with the conditional distributions introduced in [4, 10].

**Lemma 8.** *Under the notations and assumptions of Lemma 7 there are probability measures $P_{a,0}, P_{a,1}, \ldots, P_{a,M_\epsilon}$ on $E_X \times E_Y^a$ each with marginal distribution $\pi$ on $E_X$, for which the Bayes estimators $f_{*,i}^a$, $i = 1, \ldots, M_\epsilon$ satisfy $f_{*,i}^a = f_i + r$, $r \in \mathbb{R}$ where the $f_i's$ have been introduced in Lemma 7. Furthermore,*

$$KL(P_{a,i}, P_{a,j}) \leq 40B_\infty^2 C^\gamma \epsilon m^{-\gamma/p}$$

*for all $i, j \in \{0, \ldots, M_\varepsilon\}$ with $i \neq j$, where $KL$ denotes the Kullback-Leibler divergence and $C, B_\infty$ come from Lemma 7.*

*Proof.* For all $i = 1, \ldots, M_\epsilon$, recall that $\|f_i\|_{L_\infty(\pi)} \leq B_\infty$. Pick any point $r \in \mathbb{R}$ such that $r - L$ and $r + L$ belong to $E_Y^a$ where $L := 1.5 B_\infty$. Define the joint distribution $P_{a,i}(x, y_a) = p_{a,i}(y_a \mid x)\pi(x)$ where

$$p_{a,i}(y_a \mid x) = \frac{1}{2L}\left\{(L - f_i(x))\delta_{r-L}(\{y_a\}) + (L + f_i(x))\delta_{r+L}(\{y_a\})\right\}, \quad y_a \in E_Y^a \quad (39)$$

where $\delta_{r\pm L}$ is a Dirac measure on $E_Y^a$ at point $r \pm L$. $p_{a,i}(. \mid x)$ defines a probability distribution on $\mathbb{R}$ such that

$$f_{*,i}^a(x) = \int_\mathbb{R} y_a dp(y_a \mid x) = \frac{1}{2L}\left\{(L - f_i(x))(r - L) + (L + f_i(x))(r + L)\right\} = r + f_i(x).$$

We now investigate the KL divergence between $P_{a,i}$ and $P_{a,j}$. The proof is the same as in Proposition 4 of [4] and Lemma 3.2 of [10]. We first note that

$$\log\left(\frac{L \pm f_i(x)}{L \pm f_j(x)}\right) = \log\left(1 \pm \frac{f_i(x) - f_j(x)}{L \pm f_j(x)}\right)$$
$$\leq \pm\frac{f_j(x) - f_j(x)}{L \pm f_j(x)}.$$

Therefore, we can bound the KL divergence between $P_{a,i}$ and $P_{a,j}$ as

$$KL(P_{a,i}, P_{a,j}) \leq \frac{1}{2L}\int_{E_X} \frac{f_i(x) - f_j(x)}{L + f_j(x)}(L + f_i(x)) - \frac{f_i(x) - f_j(x)}{L - f_j(x)}(L - f_i(x))d\pi(x)$$
$$= \frac{1}{2L}\int_{E_X} \frac{f_i(x) - f_j(x)}{L + f_j(x)}(L + f_j(x) + f_i(x) - f_j(x))$$
$$- \frac{f_i(x) - f_j(x)}{L - f_j(x)}(L - f_j(x) + f_j(x) - f_i(x))d\pi(x)$$
$$= \frac{1}{2L}\int_{E_X} \frac{(f_i(x) - f_j(x))^2}{L + f_j(x)} + \frac{(f_i(x) - f_j(x))^2}{L - f_j(x)}d\pi(x)$$
$$= \int_{E_X} \frac{(f_i(x) - f_j(x))^2}{L^2 - f_i^2(x)}d\pi(x) \leq 1.25 B_\infty^2 \|f_i - f_j\|_{L_2(\pi)}^2$$
$$\leq 40 B_\infty^2 C^\gamma \epsilon m^{-\gamma/p}.$$

$\square$

Combining Lemma 7 and Lemma 8 allows us to derive a lower bound on the scalar-valued regression associated to $D_a$. The proof of the following Theorem is a consequence of Theorem 20, Lemma 19 and Theorem 2 in [11].

**Theorem 5.** *Under the notations and assumptions of Lemma 7 there exists constants $K_0, K, s > 0$ such that for all learning methods $D_a \to \hat{f}_{D_a}$, all $\tau > 0$, and all sufficiently large $n \geq 1$ there is a distribution $P_a$ defined on $E_X \times E_Y^a$ used to sample $D_a$, with marginal distribution $\pi$ on $E_X$ such that $f_{a,*} \in [\mathcal{H}]_X^\beta$, $\|f_{a,*}\|_\beta \leq B$, and $\|f_{a,*}\|_\infty \leq B_\infty$, and with probability not less than $1 - K_0\tau^{1/s}$,*

$$\|[\hat{f}_{D_a}] - f_{a,*}\|_\gamma^2 \geq \tau^2 K n^{-\frac{\max\{\alpha,\beta\}-\gamma}{\max\{\alpha,\beta\}+p}}.$$

We now use Theorem 5 in conjunction with Eq. (35) to prove Theorem 3.

*Proof of Theorem 3.* The conditional distribution used in the proof of Lemma 8 Eq. (39) to obtain a lower bound on the scalar-valued regression risk takes the form

$$p_a(y_a \mid x) = \frac{1}{2L}\left\{(L - f(x))\delta_{r-L}(\{y_a\}) + (L + f(x))\delta_{r+L}(\{y_a\})\right\} 1_{y_a \in \phi_Y(E_Y)(a)}, \quad y_a \in \mathbb{R} \quad (40)$$

with $L = 1.25 B_\infty$, $f \in [\mathcal{H}]_X^\beta$, $\|f_\omega\|_\beta \le \bar{B}$ and $\|f_\omega\|_{L_\infty(\pi)} \le B_\infty$. Since $r \pm L \in E_Y^a$, there exists $y_\pm \in E_Y$ such that $\phi_y(y_\pm)(a) = r \pm L$. Therefore, for all $x \in E_X$,

$$p(y \mid x) = \frac{1}{2L} \left\{ (L - f(x))\delta_{y_-}(\{y\}) + (L + f(x))\delta_{y_+}(\{y\}) \right\}, \qquad y \in E_Y \tag{41}$$

defines a family of contional distributions on $E_Y$ such that $p_a(. \mid x) = (k_Y(\cdot, a))_\# \, p(\cdot \mid x)$. For the joint distribution $p(x, y) = p(y \mid x)\pi(x)$ the conditional mean embedding is

$$\begin{aligned} F_*(x) &= \int_{E_Y} \phi_Y(y) dp(y \mid x) \\ &= \frac{1}{2L} \left\{ (L - f(x))\phi_Y(y_-) + (L + f(x))\phi_Y(y_+) \right\} \end{aligned} \tag{42}$$

As a result, we have

$$\begin{aligned} \|F_*\|_\beta &= \frac{1}{2L} \left\| \{(L - f) \otimes \phi_Y(y_-) + (L + f) \otimes \phi_Y(y_+) \} \right\|_\beta \\ &\le \frac{1}{2L} \left( \|(L - f) \otimes \phi_Y(y_-)\|_\beta + \|(L + f) \otimes \phi_Y(y_+)\|_\beta \right) \\ &= \frac{1}{2L} \left( \|L - f\|_\beta \|\phi_Y(y_-)\|_{\mathcal{H}_Y} + \|L + f\|_\beta \|\phi_Y(y_+)\|_{\mathcal{H}_Y} \right) \\ &\le \frac{\kappa_Y}{2L} \left( \|L - f\|_\beta + \|L + f\|_\beta \right) \\ &= \kappa_Y + \frac{\kappa_Y}{L} \|f\|_\beta < +\infty, \end{aligned}$$

where the third step follows from Definition 3, the fourth step is due to the boundedness of kernel $k_Y$ and the second last step follows from Eq. (8). We conclude by combining Theorem 5 with Eq. (35).

$\square$

## C  Auxiliary Results

The following lemma is from [11].

**Lemma 9.** *Under* (EMB) *we have*

$$\left\| (C_{XX} + \lambda Id_{\mathcal{H}_X})^{-\frac{1}{2}} k(X, \cdot) \right\|_{\mathcal{H}_X} \le A\lambda^{-\frac{\alpha}{2}}.$$

The following Theorem is from [11, Theorem 26].

**Theorem 6.** *Bernstein's Inequality. Let $(\Omega, \mathcal{B}, P)$ be a probability space, $H$ be a separable Hilbert space, and $\xi : \Omega \to H$ be a random variable with*

$$\mathbb{E}_P \|\xi\|_H^m \le \frac{1}{2} m! \sigma^2 L^{m-2}$$

*for all $m \ge 2$. Then, for $\tau \ge 1$ and $n \ge 1$, the following concentration inequality is satisfied*

$$P^n \left( (\omega_1, \ldots, \omega_n) \in \Omega^n : \left\| \frac{1}{n} \sum_{i=1}^n \xi(\omega_i) - \mathbb{E}_P \xi \right\|_H^2 \ge 32 \frac{\tau^2}{n} \left( \sigma^2 + \frac{L^2}{n} \right) \right) \le 2e^{-\tau}$$

**Lemma 10.** *Suppose* (EVD) *holds. Then, there exists a $c > 0$ such that:*

$$\mathcal{N}(\lambda) = Tr\left( C_{XX} (C_{XX} + \lambda Id_{\mathcal{H}_X})^{-1} \right) \le c\lambda^{-p}$$

## D  Well-specifiedness of the CME problem and discussion of some corner cases

As the CME has been redefined various times, the conditions ensuring the existence of a closed-form solution have been subject to various modifications. The purpose of this section is to briefly

investigate the well-specifiedness assumptions in the operator-theoretic setting [37, 18] and in our kernel regression setting [29]. The connections between these assumptions are rather complex (we also refer to Section 5 of [19] and to Section 2.4 of [41]).

To recapitulate, well-specifiedness in the original operator-theoretic setting usually involves the requirement

$$\mathbb{E}[g(Y)|X = \cdot] \in \mathcal{H}_X \text{ for all } g \in \mathcal{H}_Y, \tag{a}$$

while well-specifiedness in the kernel regression setting means that a representative of the $L_2$-function class associated with the CME function is contained in the hypothesis space $\mathcal{G}$, which we write for simplicity as

$$\mathbb{E}[\phi_Y(Y)|X = \cdot] \in \mathcal{G}. \tag{b}$$

Before we discuss some corner cases, we first point out that condition (b) implies condition (a). To see this, we notice that by Corollary 1, we have $\mathbb{E}[\phi_Y(Y)|X = \cdot] = C\phi_X(\cdot)$ for some $C \in S_2(\mathcal{H}_X, \mathcal{H}_Y)$. Therefore, for any $g \in \mathcal{H}_Y$, we have

$$\mathbb{E}[g(Y)|X = \cdot] = \langle g, C\phi_X(\cdot)\rangle_{\mathcal{H}_Y} = \langle C^*g, \phi_X(\cdot)\rangle_{\mathcal{H}_X}.$$

It is easy to see that $C^*g \in \mathcal{H}_X$ for any $g \in \mathcal{H}_Y$, hence condition (a) is satisfied.

$Y = X$: This is an example that condition (a) does not imply condition (b). Let $\mathcal{G}$ be the vRKHS induced by the kernel

$$K(x, x') = k_X(x, x')\text{Id}_{\mathcal{H}_Y}, x, x' \in E.$$

The first example is the special case where we have $k_Y = k_X$ as well as $X = Y$. It is easy to see that this reduces the CME to

$$\mathbb{E}[\phi_X(X)|X = \cdot] = \phi_X(\cdot).$$

We can also verify that condition (a) is satisfied in this case, as we have $\mathbb{E}[g(X)|X = \cdot] = g(\cdot) \in \mathcal{H}_X$.

Furthermore, it is clear that the identity operator $\text{Id}_{\mathcal{H}_X}$ is the correct operator-theoretic solution to the CME problem, as it represents the CME in terms of $\phi_X(\cdot) = \text{Id}_{\mathcal{H}_X}\phi_X(\cdot)$. However, if $\mathcal{H}_X$ is infinite dimensional, it is also clear that $\text{Id}_{\mathcal{H}_X}$ is not Hilbert–Schmidt and hence

$$\text{Id}_{\mathcal{H}_X}\phi_X(\cdot) \notin \mathcal{G} \simeq \mathcal{H}_X \otimes \mathcal{H}_X.$$

Hence, according to condition (a), we have a well-specified setting, while according to condition (b), we clearly have a misspecified setting.

This example demonstrates that, without additional requirements, the well-specifiedness condition (a) allows cases where the CME is represented by a bounded operator, while condition (b) restricts the class of admissible representative operators to the Hilbert–Schmidt class.

$Y \perp X$: In this case, it is clear that

$$\mathbb{E}[\phi_Y(Y)|X = \cdot] = \int_{E_Y} \phi_Y(y)d\nu(y) = \mu_Y.$$

Similar to the previous case, neither condition (a) nor (b) are satisfied. Moreover, requiring that the CME is contained in $[\mathcal{G}]^\beta$ amounts to require that $\mathbb{E}[g(Y)|X = \cdot] \in [\mathcal{H}]_X^\beta$. However, when $Y$ is independent of $X$, we have $\mathbb{E}[g(Y)|X = \cdot] = \mathbb{E}[g(Y)]$ which is a constant function. Since the constant function is included in $[\mathcal{H}]_X^\beta$ for $\beta = 0$, essentially independence between $Y$ and $X$ is equivalent to the case where the target CME is contained in $[\mathcal{G}]^0$.