# OpenReview forum: "Optimal Rates for Regularized Conditional Mean Embedding Learning"
_NeurIPS.cc/2022/Conference — NeurIPS 2022 Accept_

### Official Review · Reviewer_x4Bu · 2022-07-11

**Rating:** 7
**Confidence:** 2
**Soundness:** 2 fair
**Presentation:** 2 fair
**Contribution:** 2 fair

**Summary:**

The paper considers estimating the conditional mean embedding (CME). The authors propose an estimation of CME using an approximation in interpolation space.  Consistency and convergence rates are also established for the CME estimator in the misspecified setting.


**Questions:**

See the main comments.

**Strengths And Weaknesses:**

[Strengths]
1. The presentation of the paper is clear, and estimating the CME is an important problem.
2. The authors provide an operator-theoretic definition of the CME in the misspecified case where the target CME can be well approximated by Hilbert-Schmidt operators.
3. A lower bound on the learning rate is established, which reveals that when the target CME operator is smooth, the obtained upper bound is optimal.

[Weaknesses]
1. It seems that most of the results follow from [8, 39] as well as direct adaption of the interpolation spaces. The background section could be reduced.
2. I am not convinced about the advantages of adapting the interpolation space for approximation. How does the proposed approach compare with related work [39]?
3. While most of the work is theoretical, it could be useful to provide numerical results to demonstrate the performance on real-life data.

---

> ### Author Response · Authors · 2022-08-02
> **Response**
>
> We thank the reviewer for their time and effort in reading our manuscript. We agree that certain aspects of the analysis make use of techniques from Fischer and Steinwarts (2020) and Talwai et al., (2022), and we have been careful in citing these earlier results where they are used.
>
> There are several key steps that differentiate our paper from Fischer and Steinwarts (2020) and Talwai et al., (2022), which we hope we have made clearer in our revised submission:
>
> First, a real-valued interpolation space is defined in Fischer and Steinwarts (2020), in which the misspecified results were studied. However, the CME requires a vector-valued interpolation space, which we show is isometric to the Hilbert-Schmidt operator space. This allows us to define in a systematic way what it means to be well-specified and misspecified in the case where the output is itself an RKHS, rather than a scalar as in Fischer and Steinwarts (2020) . This then required significant changes to the proofs of Fischer and Steinwarts (2020) , as detailed in the paper.
>
> In relation to Talwai et al., (2022), we have added a new remark 5, which replaces the discussion at the end of p. 8 in our original submission. There are two important points that distinguish our work: first, our construction avoids the $\beta > p$ requirement of Talwai et al., (2022): this limitation meant that the earlier proof was restricted to “near-well-specified” settings even for reasonable choices of kernel (see the remark for a more rigorous discussion). Second, we provide a lower bound, which establishes that our result is minimax optimal. No such proof was provided in Talwai et al., (2022).
>
> In respect of numerical applications of conditional mean embeddings, these may be found in some of the cited work, for instance Gr{\"u}new{\"a}lder et al., (2012), Sejdinovic et al., (2013), Song et al., (2013), Song et al., (2010), Song et al., (2009). We focus here on providing a theoretical foundation to support these empirical studies, as was done for the scalar case in Fischer and Steinwart (2020).

---

### Official Review · Reviewer_u4KM · 2022-07-12

**Rating:** 7
**Confidence:** 4
**Soundness:** 4 excellent
**Presentation:** 3 good
**Contribution:** 4 excellent

**Summary:**

This paper presents novel theoretical results on learning empirical approximations of conditional mean embeddings (CMEs) from samples. The derivations focus on the misspecified case, where the true CME operator does not satisfy the usually strict assumptions found in the literature, but only milder ones which are more plausible in practical settings. The theoretical analysis show that the true CME can be approximated via an interpolation space between the reproducing kernel Hilbert space $\mathcal{H}_X$ of functions over the conditioning inputs $X$ and the Hilbert space $L_2(\pi)$ of square-integrable functions under the inputs marginal distribution $X \sim \pi$.The derived results present an optimal learning rate for the empirical CME approximation, which leads to a vanishing error bound with respect to the true CME operator.

**Questions:**

* I've missed some important details:
 - It is not specified whether the samples in the dataset for learning CME estimator need to be independent and identically distributed (i.i.d.) w.r.t. the joint the distribution of $X, Y$ or not. Would the results hold for a non-i.i.d. case?
- The statement of Theorem 2 is a bit confusing regarding $\alpha$. Does Thm. 2 hold for any $\alpha > 0$? Or does it imply there is an $alpha > 0$ such that its result holds?
- It is not clear to me how to get to the formulation of $C_{XX}$ in terms of $(\mu_i)_i$ and $(e_i)_i$ in line 135.

* Minor details:
 - In the proof sketch after Theorem 4, are we missing an identity operator next to $\lambda$ in the definition of $F_{Y|X,\lambda}$?


**Limitations:**

Most limitations are discussed in the paper.

**Strengths And Weaknesses:**

Strengths:
- The theoretical analysis is rigorous and improves on the understanding of CME learning in misspecified settings.
- Assumptions are milder and seem easier to satisfy than the ones found in existing theoretical analyses in the literature.
- The first lower bound on the learning rate for the CME estimator is derived.

Weaknesses:
- The derived results do not hold for the case of the constant function, i.e., when $Y$ is independent of $X$, which is a problem in the case of radial basis function and other characteristic kernels popular in kernel embeddings applications.
- The paper is quite dense at some points and a few details get lost.

---

> ### Author Response · Authors · 2022-08-02
> **Response**
>
> We thank the reviewer’s time and effort in providing such encouraging feedback. We address the questions raised by the reviewer below.
>
> For the problem of constant functions, we have included a new discussion in the updated draft (Remark 6) and accompanying Appendix D. This remark discusses the case where $Y$ is independent of $X$ and shows that the constant function does not belong to the space $[\mathcal{G}]^{\beta}$ unless $\beta = 0$, hence consistency in this corner case indeed remains to be established (as we now clarify).
>
> Regarding the denseness of the paper, we have updated our manuscript with the aim of improving clarity. We believe the revised manuscript is easier to follow, but we welcome further suggestions for improvement.
>
> Regarding the questions raised by the reviewer:
> In our study, the samples are indeed required to be i.i.d from the joint distribution of $X$ and $Y$. There is interesting work for non-i.i.d. data in the regression setting, see for instance: Hang, Steinwart, “Fast learning from alpha-mixing observations,” Journal of Multivariate Analysis, Volume 127, May 2014, Pages 184-199. The generalization of our result to the non-i.i.d. case will be the focus of future study.
> Re Theorem 2: this theorem is no longer needed in the updated manuscript.
> C_XX can be derived in the following way
> $$C_{XX} = \mathbb{E}(\phi_X(x) \otimes \phi_X(x)) = \mathbb{E}( \sum_i \mu_i e_i(x)e_i(\cdot) \otimes \sum_j \mu_j e_j(x)e_j(\cdot) )  = \sum_i \mu_i \mu_i^{1/2} e_i(\cdot) \otimes \mu_i^{1/2} e_i(\cdot).$$
> Thanks for the comment, we have addressed the typo.
>
> Finally, please see the notes above for revisions and improvements we’ve now made to our manuscript.

---

> > ### Comment · Reviewer_u4KM · 2022-08-06
> > **Acknowledgement**
> >
> > I'd like to thank the authors for addressing mine and other reviewers' concerns. I believe the paper makes a significant contribution to the theory of kernel mean embeddings in misspecified settings and should be accepted.

---

### Official Review · Reviewer_AJfs · 2022-07-12

**Rating:** 7
**Confidence:** 3
**Soundness:** 4 excellent
**Presentation:** 4 excellent
**Contribution:** 3 good

**Summary:**

The work focuses on estimation of the kernel conditional embedding operator in the so-called mis specified case. The recently proposed idea of interpolation spaces between H_X and L_2 is leveraged to define CME operator with milder conditions (depending on interpolant parameter \beta).

The key contribution is theorem 4, where an upper bound on estimation error of the CME operator is presented (mis specified case). Though the derivation is technical, the final bound is insightful and depends on the interplay between \beta, \apha (smoothness parameter) and p (eigen value decay parameter). For larger \beta (i.e., less mis specification), in particular, \alpha\le\beta, which is shown to be optimal via theorem 5 that provides a matching lower bound. The upper bound however can be very loose when \beta is tiny.



**Questions:**

Please verify minor typos:
Consider five equations between eqn(14) and line 479. The fourth equation misses a C_YX term. But it is minor because it is back in the next equation.

**Limitations:**

Limitations have been discussed (case of low \beta).

**Strengths And Weaknesses:**

Overall the paper is well-written and organized. Though technical, enough details are provided to be self-contained. The upper bound is interesting as it explicilty involves the mis-specification parameter. The lower bound on learning rate is also seems to fill a gap in existing literature.

---

> ### Author Response · Authors · 2022-08-02
> **Response**
>
> We thank the reviewer for their kind assessment of our manuscript. The mentioned typo is now fixed in the new version. In addition, please see the notes above for further revisions and improvements we’ve made to our original submission.

---

### Official Review · Reviewer_nfGK · 2022-07-15

**Rating:** 9
**Confidence:** 2
**Soundness:** 4 excellent
**Presentation:** 4 excellent
**Contribution:** 4 excellent

**Summary:**

This paper studies the conditional mean embedding and provides learning rate for the ridge regression based estimator of the CME.

**Questions:**

-

**Limitations:**

Not relevant

**Strengths And Weaknesses:**

This is a very technical paper and as such I have not verified all the details. Despite being technical the paper does a great job in being clear and precise about the statements.

Conditional mean embeddings (CME), although used frequently in machine learning, have only recently been defined and studied rigorously. This paper is part of that literature and it makes some valuable contributions. The most important of them is using interpolation space theory to study an estimator for the CME operator and provide optimal learning rates.

---

> ### Author Response · Authors · 2022-08-02
> **Response**
>
> We thank the reviewer for their encouraging comments. Please see the notes above for revisions and improvements we’ve now made to our manuscript.

---

### Author Response · Authors · 2022-08-02
**Updates**

Dear Reviewers,

We thank you for your time and effort in reviewing our manuscript, and sincerely appreciate your encouraging feedback and insightful comments.  We reply to reviewer questions in individual responses below.

We have updated our manuscript, incorporating changes made in response to reviewer suggestions,  and with additional improvements that  broaden the scope of the results. A high level description of the latter updates is as follows:

First, we have extended our analysis of the upper bound from the vector-valued interpolation RKHS $\mathcal{G}^{\beta}$ (which maps from an “enlarged” RKHS) to the vector-valued interpolation space $[\mathcal{G}]^{\beta}$ (which maps from subspaces of $L_2$). The proofs follow largely as before (in fact, the Bernstein conditions for the variance proof are now a bit simpler!), however this is more than a cosmetic update, since in certain cases $\mathcal{G}^{\beta}$ might not exist. Thus the new result is strictly more general. Please refer to Remark 5 in the updated manuscript.

Second, we discovered that our original proof for the lower bound makes an implicit restrictive requirement on the RKHS $\mathcal{H}_Y$, which is that the required Gaussian distributions on  $\mathcal{H}_Y$ must be representable by distributions on  $E_Y$. This requirement arises on line 522 of the original supplementary material. Unfortunately, while the construction is possible in some RKHS  $\mathcal{H}_Y$ it may not hold across all of the infinite dimensional RKHSs $\mathcal{H}_Y$ that interest us, notably the RKHS induced by the exponentiated quadratic kernel.

Fortunately, we found that there is a far easier way to get the required lower bound. We simply perform a scalar regression to $k(y,a)$, rather than directly addressing the problem of predicting the potentially infinite dimensional $\phi_Y(y)$. This straightforwardly gives a lower bound on the original problem (eq. 33 in the revised submission),  which can then be addressed using the scalar regression results of Fischer and Steinwart (2020) and Caponnetto and De Vito (2007). In the revised submission, we cite all the results we need, since many steps now carry across with little or no modification once the original step of eq. 33 has been made.

We hope these revisions will strengthen the paper, and we’d be happy to clarify any further points that might arise.

---

### Meta-Review · Area_Chair_1JKX · 2022-08-26

**Recommendation:** Accept
**Confidence:** Certain

**Metareview:**

There is a wide consensus among reviewers, after the discussion period, that this submission has strong and novel results where a clear minimax optimality is established for the conditional mean embedding problem.

For the camera ready version, the authors are strongly encouraged to clearly implement all revisions mentioned in the discussion related to the correctness of the lower bound.

**Award:**

No

---

### Decision · Program_Chairs · 2022-09-14

Accept